# Distinct mechanisms control the specific synaptic functions of Neuroligin 1 and Neuroligin 2

Jinzhao Wang [1,2,3,4], Thomas Sudhof [1,2,5✉] & Marius Wernig [1,3,4✉]

## Abstract

**Neuroligins are postsynaptic cell-adhesion molecules that regulate synaptic function with a remarkable isoform specificity. Although Nlgn1 and Nlgn2 are highly homologous and biochemically interact with the same extra- and intracellular proteins, Nlgn1 selectively functions in excitatory synapses whereas Nlgn2 functions in inhibitory synapses. How this excitatory/inhibitory (E/I) specificity arises is unknown. Using a comprehensive structure-function analysis, we here expressed wild-type and mutant neuroligins in functional rescue experiments in cultured hippocampal neurons lacking all endogenous neuroligins. Electrophysiology confirmed that Nlgn1 and Nlgn2 selectively restored excitatory and inhibitory synaptic transmission, respectively, in neuroligin-deficient neurons, aligned with their synaptic localizations. Chimeric Nlgn1-Nlgn2 constructs reveal that the extracellular neuroligin domains confer synapse specificity, whereas their intracellular sequences are exchangeable. However, the cytoplasmic sequences of Nlgn2, including its Gephyrin-binding motif that is identically present in the Nlgn1, is essential for its synaptic function whereas they are dispensable for Nlgn1. These results demonstrate that although the excitatory vs. inhibitory synapse specificity of Nlgn1 and Nlgn2 are both determined by their extracellular sequences, these neuroligins enable normal synaptic connections via distinct intracellular mechanisms.**

**Keywords** Neuroligin; Synapse Specific; Extracellular Domain; Intracellular Domain; Distinct Mechanisms
**Subject Category** Neuroscience

## Introduction

Neuroligins (Nlgns) are evolutionarily conserved postsynaptic cell adhesion molecules that bind to pre-synaptic neurexins (Nrxns) (Ichtchenko et al, 1995; Ichtchenko et al, 1996; Nguyen and Sudhof, 1997). In vertebrates, four genes encode Nlgns: Nlgn1, Nlgn2, Nlgn3, and Nlgn4. Among these, Nlgn1, Nlgn2, and Nlgn3 exhibit high conservation across species, while Nlgn4 displays variability between rodents and humans and is expressed at relatively low levels in mice (Bolliger et al, 2008; Ichtchenko et al, 1995; Ichtchenko et al, 1996; Jamain et al, 2003). Nlgns are type 1 transmembrane proteins that comprise a single, large extracellular domain consisting of a constitutively dimeric, enzymatically inactive esterase-homology domain, a transmembrane region (TMR), and a short cytoplasmic tail. Remarkably, the targeting of Nlgns to synapses is selective and specific. Nlgn1 is only localized to glutamatergic synapses (Song et al, 1999) while Nlgn2 is exclusively present at GABAergic, dopaminergic, and cholinergic synapses (Graf et al, 2004; Takacs et al, 2013; Uchigashima et al, 2016; Varoqueaux et al, 2004). Nlgn3 is found in both excitatory and inhibitory synapses (Budreck and Scheiffele, 2007), and Nlgn4 functions in glycinergic synapses in mice and in GABAergic synapses in human neurons (Hoon et al, 2011; Marro et al, 2019; Zhang et al, 2018). Consistent with their localization to synapses, Nlgns perform multiple synapse-specific functions. Compelling initial evidence supporting the synaptic function of Nlgns was obtained through overexpression experiments, which demonstrated significant increases in synapse density and synaptic transmission in transfected neurons (Chubykin et al, 2007; Scheiffele et al, 2000). Since then, a large number of functional studies in invertebrates and vertebrates using both deletions of and mutations in Nlgn genes revealed a panoply of important roles for Nlgns in shaping synapses, confirming the central contribution of Nlgns to the functional architecture of synaptic connections (Banovic et al, 2010; Calahorro and Ruiz-Rubio, 2012; Chubykin et al, 2007; Hu et al, 2012; Hunter et al, 2010; Maglioni et al, 2022; Sudhof, 2008; Sun et al, 2011; Sun et al, 2023; Tabuchi et al, 2007; Tu et al, 2015; Varoqueaux et al, 2006; Xing et al, 2014).

Despite much investigation, however, several key questions about Nlgn function remain unaddressed. Possibly the most important of these questions regards the observation that different Nlgn isoforms, despite a high degree of sequence similarity, perform largely non-overlapping distinct functions in the same neurons. How are different Nlgns targeted to distinct synapses where they perform different roles? This question represents an intriguing cell biological challenge given that Nlgns are post-synaptic, and each neuron may receive inputs from thousands of neurons using the whole range of neurotransmitter types. Since Nlgns interact with synapse-specific molecules at both the

[1]Institute for Stem Cell Biology and Regenerative Medicine, Stanford University School of Medicine, Stanford, CA 94305, USA. [2]Department of Molecular and Cellular Physiology, Stanford University School of Medicine, Stanford, CA 94305, USA. [3]Department of Pathology, Stanford University School of Medicine, Stanford, CA 94305, USA. [4]Department of Chemical and Systems Biology, Stanford University School of Medicine, Stanford, CA 94305, USA. [5]Howard Hughes Medical Institute, Stanford University School of Medicine, Stanford, CA 94305, USA. ✉E-mail: tcs1@stanford.edu; wernig@stanford.edu

post-synaptic (on the same cell) and pre-synaptic side (with molecules from the projecting neuron), E/I specificity could be conferred to various Nlgns by intra- and/or extra-cellular mechanisms. For example, Nlgn2 is primarily localized to inhibitory synapses and Nlgn2 loss-of-function results in impaired inhibitory but not excitatory synaptic transmission (Chanda et al, 2017; Chubykin et al, 2007; Gibson et al, 2009; Liang et al, 2015; Poulopoulos et al, 2009; Zhang et al, 2015). Intracellularly, Nlgn2 binds to the inhibitory synapse-specific molecules gephyrin, collybistin, GARLH3, and GARLH4, which are plausible candidates to recruit Nlgn2 to inhibitory synapses (Poulopoulos et al, 2009; Yamasaki et al, 2017). However, the intracellular domains of different Nlgns are highly conserved, which argues against a strong contribution of intracellular interaction partners as a mechanism of E/I specificity. On the extracellular side facing pre-synaptic terminals, several molecules are also known to bind to Nlgns. The best characterized Nlgn interacting proteins are the three Nrxns (Nrxn1-3), each of which is expressed in hundreds of alternatively spliced variants (Treutlein et al, 2014). With this impressive possibility of combinations, it is conceivable that specific "Nrxn codes" could decorate different classes of synapses (Aoto et al, 2015; Dai et al, 2019; Dai et al, 2021). More recently, the extracellular MDGA proteins have been described as Nlgn regulators by binding to Nlgns in competition with Nrxns (Connor et al, 2019; Lee et al, 2013; Tanaka et al, 2012). Crystal structures of MDGA1/Nlgn1 and MDGA1/Nlgn2 complexes revealed that two N-terminal lg domains of MDGA1 straddle the Nlgn1 or Nlgn2 homodimer, such that each Nlgn homodimer binds to two MDGA1 molecules (Connor et al, 2016; Gangwar et al, 2017; Kim et al, 2017; Lee et al, 2013). The binding of Nlgns to MDGAs has been hypothesized to restrain Nlgn functions (Elegheert et al, 2017b; Pettem et al, 2013), but much about the mechanisms and relative importance of this interaction is unknown. In summary, how Nlgns target and exert their functions at specific types of synapses remains a mystery.

Here, we begin to address some of these critical questions using expression of specific Nlgns and their mutants in cultured hippocampal neurons in which all Nlgn isoforms were genetically deleted, with functional analyses performed by electrophysiology and imaging. This approach reduces the complexity of potential interactions among different Nlgn isoforms that can be present in the same postsynaptic compartment and elucidates the inherent functions of the single Nlgn molecules expressed in this system, thereby simplifying functional interpretations.

## Results

### Specific synaptic targeting is an inherent property of Nlgn1 and Nlgn2

The co-existence of various Nlgns in synapses complicates the conclusions about a particular Nlgn using conventional experimental approaches. We therefore sought to reduce the complexity to a neuronal cell system that contains only a single Nlgn of interest. To this end, we generated quadruple conditional knock-out (cKO) mice where all 4 endogenous Nlgn genes are conditionally deleted upon expression of Cre recombinase. Our experimental setup is illustrated in Fig. 1A and entails (i) the

generation of primary hippocampal cultures of postnatal day 0 (P0) mice, (ii) the expression by lentiviral transduction of active Cre recombinase or inactive ΔCre recombinase as a control, and (iii) the co-expression of a specific Nlgn construct. We performed immunoblots on lysates from Nlgn1234 cKO hippocampal cultures infected with lentiviral ΔCre or Cre (Fig. EV1A,B). Cre expression caused a substantial loss of Nlgn1, Nlgn2, and Nlgn3 (~70–75%). Nlgn4 is not significantly expressed in this culture system (Chanda et al, 2017; Hoon et al, 2011; Varoqueaux et al, 2006).

To investigate the localization of Nlgn1 and Nlgn2 (that are known to localize to specifically to excitatory and inhibitory synapses, respectively (Graf et al, 2004; Poulopoulos et al, 2009; Song et al, 1999; Varoqueaux et al, 2004)), in the absence of any other Nlgns, we expressed N-terminally hemagglutinin (HA)-tagged, full-length Nlgn1 or Nlgn2 under control of the Synapsin promoter in hippocampal neurons after Cre-mediated deletion of all endogenous Nlgns (Fig. 1B). Co-labeling with excitatory and inhibitory synapse markers allowed us to assess the specific targeting of Nlgn1 and Nlgn2 by fluorescence imaging. We detected strong expression of the infected Nlgn1 and Nlgn2 on MAP2-positive dendrites in a punctate pattern (Fig. 1C–F). We also co-stained HA with Homer1 or Gephyrin as markers of excitatory and inhibitory synapses, respectively (Fig. 1C,E). Consistent with reports about endogenous Nlgn1 and Nlgn2 (Song et al, 1999; Varoqueaux et al, 2004), we found that in the absence of endogenous Nlgns, the majority of HA-Nlgn1 puncta were co-localized with Homer1, representing ~60% of the observed puncta. In contrast, HA-Nlgn1 puncta showed little co-localization with Gephyrin, accounting for only ~27% of the observed puncta. HA-Nlgn2 puncta, conversely, predominantly co-localized with Gephyrin, with ~59% of the puncta showing this co-localization. Only ~28% of HA-Nlgn2 puncta co-localized with Homer1 (Fig. 1D,F). We also analyzed the co-localization of Homer1 and Gephyrin to address synapse specificity at the confocal level. We found ~21% Homer1 co-localization with Gephyrin and ~30% Gephyrin co-localization with Homer1 (Fig. EV1C,D). However, the background of positive co-localization with the 'wrong' synapse is high due to the insufficient imaging resolution for separating closely spaced synapses and the significant impact of thresholding on apparent colocalizations. Although these findings are limited by the resolution of confocal microscopy, they are consistent with previous conclusions that Nlgn1 primarily targets excitatory synapses while Nlgn2 primarily targets inhibitory synapses, showing that Nlgn1 and Nlgn2 exhibit this specificity even in the absence of other Nlgns.

### The Gephyrin-binding domain of Nlgn2 is necessary for its inhibitory synapse-specific function but does not confer inhibitory synapse specificity

To investigate the molecular mechanisms targeting Nlgn2 to inhibitory synapses, we generated Nlgn2 constructs with a truncation of the intracellular domain (Nlgn2-GPI) and a chimeric Nlgn2 construct whose intracellular domain was switched with the corresponding Nlgn1 sequence (Fig. 2A). We then expressed the constructs in our Nlgn1234 cKO hippocampal neuron system. Cell surface staining of Nlgn2-GPI and Nlgn2-Nlgn1 constructs in the Cre-infected neurons confirmed proper surface localization, indicating that the introduced mutations do not interfere with

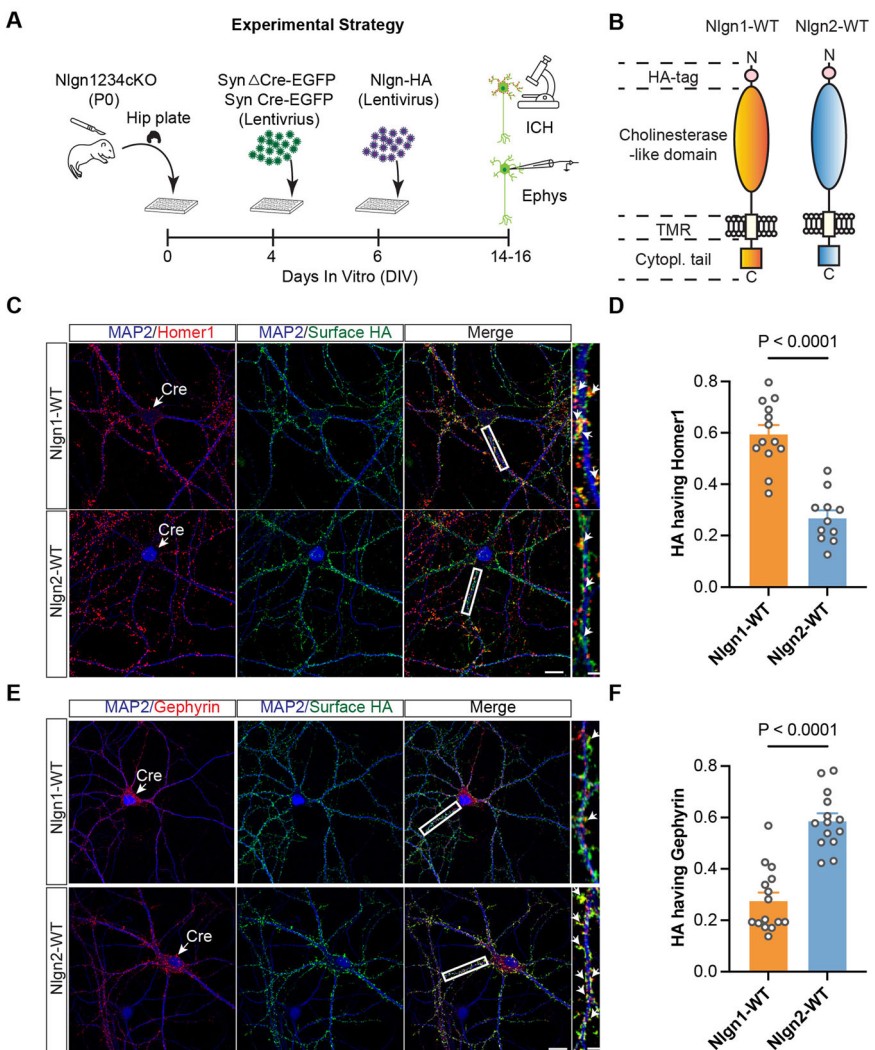

**Figure 1. Nlgn1 and Nlgn2 are specifically localized on excitatory and inhibitory synapses.**

(A) Schematic overview of experimental timeline and set-up. (B) Schematic of Nlgn1-WT and Nlgn2-WT constructs. (C) Representative image from DIV14–16 cultured Nlgn1234 conditional knockout mice neurons, infected with Cre (blue) and either Nlgn1-WT (top) or Nlgn2-WT (bottom). The neurons were labeled with antibodies to Homer1 (red), HA (green), and MAP2 (blue). Scale bar: 20 µm. The right panels show an enlarged box area (arrowheads indicate HA puncta overlapped with Homer1 puncta). Scale bar: 5 µm. (D) Summary graph of the HA-Homer1 overlap percentage in Nlgn1-WT and Nlgn2-WT conditions. (Bar and line graphs indicate mean ± SEM; numbers of cells/experiments = 13/3 and 11/3 for each column, left to right. Statistical significance was assessed by unpaired t test, ****$p < 0.0001$). (E) Representative image of hippocampal neurons from DIV14–16 cultured Nlgn1234 conditional knockout mice, infected with Cre (blue) and either Nlgn1-WT (top) or Nlgn2-WT (bottom). The neurons were labeled with antibodies to Gephyrin (red), HA (green), and MAP2 (blue). Scale bar: 20 µm. The right panels show an enlarged box area (arrowheads indicate HA puncta overlapped with Gephyrin puncta). Scale bar: 5 µm. (F) Summary graph of the HA-Gephyrin overlap percentage in Nlgn1-WT and Nlgn2-WT conditions. (Bar and line graphs indicate mean ± SEM; numbers of cells/experiments = 15/3 and 14/3 for each column, left to right. Statistical significance was assessed by unpaired t test, ****$p < 0.0001$). Source data are available online for this figure.

protein synthesis and processing in the secretory pathway (Fig. EV2A). Remarkably, co-labeling with antibodies against the inhibitory synapse protein Gephyrin revealed that both constructs localized to inhibitory synapses, but Nlgn2-Nlgn1 exhibited a higher degree of localization with Gephyrin compared to Nlgn2-GPI (Fig. EV2B), and expression levels no significant change (Fig. EV2C).

Next, we assessed the synaptic function of the Nlgn2 constructs using electrophysiology. As expected, Cre-infected Nlgn1234 cKO hippocampal neurons exhibited a ~50% decrease of evoked GABAR-mediated inhibitory postsynaptic currents (IPSC) compared to ΔCre-

EGFP infected cultures (Fig. 2B,C), consistent with previous findings (Chanda et al, 2017; Gan and Sudhof, 2020). Expression of full-length Nlgn2 resulted in a full rescue of the GABAR-mediated IPSCs, demonstrating that Nlgn2 is the main Nlgn isoform responsible for the inhibitory synapse phenotype observed with Ngln1234 deletions (Fig. 2B,C). Intriguingly, expression of Nlgn2-GPI, i.e., Nlgn2 without an intracellular domain, failed to rescue the phenotype (Fig. 2B,C), revealing that Nlgn2 function for inhibitory synapses requires an intracellular domain presumably due to the absence of synaptic targeting motifs in the GPI-anchor domain and the impact of membrane anchoring on protein trafficking and synaptic recruitment.

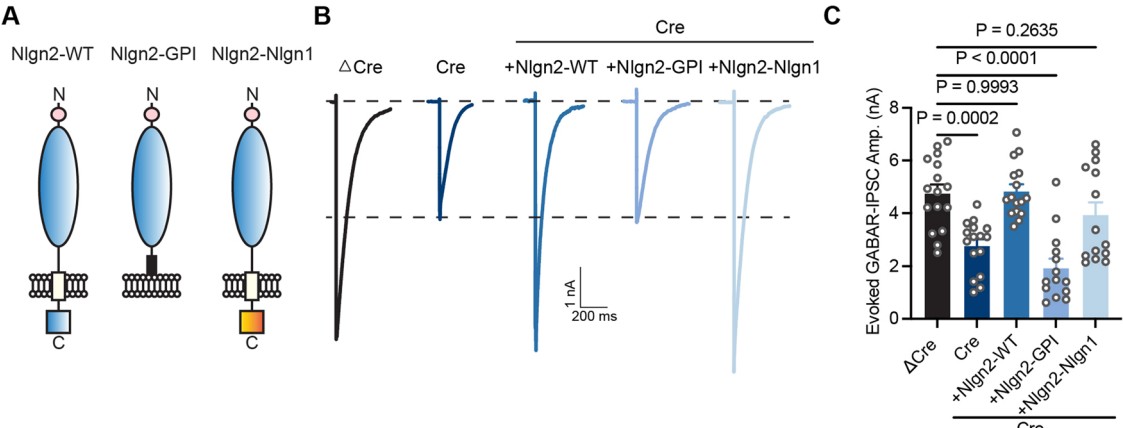

**Figure 2.** **The intracellular sequence of Nlgn2 is necessary for inhibitory synapse function but can be fully replaced with the intracellular sequence of Nlgn1.**

(A) Schematic of Nlgn2-WT, Nlgn2-GPI (Nlgn2 chimeric construct only has extracellular domain), Nlgn2-Nlgn1 (Nlgn2 extracellular domain with Nlgn1 intercellular domain) constructs. (B) Representative traces of evoked GABAR-IPSC recorded from DIV14-16 cultured Nlgn1234 conditional knockout mice neurons in the five conditions (ΔCre, Cre, Cre+Nlgn2-WT, Cre+Nlgn2-GPI, and Cre+Nlgn2-Nlgn1). (C) Summary graph of evoked GABAR-IPSC amplitude in all conditions (Bar and line graphs indicate mean ± SEM; numbers of cells/experiments = 16/3, 16/3, 16/3, 14/3, and 14/3 for each column, left to right). Nonsignificant $p > 0.05$; **$p < 0.01$; ***$p < 0.001$; ****$p < 0.0001$, one-way ANOVA with post hoc Dunnett's Multiple comparisons. Source data are available online for this figure.

Given the sequence similarity between the intracellular domains of Nlgns despite their different synaptic functions, we assessed the function of a chimeric Nlgn2 with an intracellular domain of Nlgn1. Consistent with previous results (Nguyen et al, 2016), the Nlgn2-Nlgn1 construct could also rescue the GABAR-mediated IPSC phenotype in Nlgn1234 cKO neurons (Fig. 2B,C). Thus, the intracellular Nlgn2 domain is necessary for inhibitory synapse function but does not confer specificity.

To further understand how Nlgns might function at synapses, we aligned Nlgn1 and Nlgn2 intracellular sequences and confirmed that both Nlgn1 and Nlgn2 contain a highly conserved Gephyrin-binding sequence (Fig. 3A), even though Nlgn1 is primarily localized to excitatory synapses and has no function in inhibitory synapses (Chanda et al, 2017; Song et al, 1999; Zhang et al, 2015). This raised the question whether the Gephyrin-binding sequence is responsible for the Nlgn1 and Nlgn2 intracellular domain rescue of Nlgn2 function. To address this question, we deleted the entire 15 amino acid-long Gephyrin-binding sequence from the Nlgn2-Nlgn1 construct or introduced the Y770A point mutation which was shown to prevent Nlgn2 binding to Gephyrin by blocking the phosphorylation of this residue (Giannone et al, 2013; Poulopoulos et al, 2009) (Fig. 3B). Unexpectedly, expression of Nlgn2-Nlgn1Y770A construct fully rescued the IPSC phenotype, whereas the construct with the deletion of the entire Gephyrin-binding sequence (Nlgn2-Nlgn1 DelGeph) did not rescue (Fig. 3C,D). Again, we confirmed that the expressed constructs are properly transported to the cell surface by neuronal surface staining (Fig. EV2D). In line with our electrophysiology data, co-labeling with Gephyrin antibodies showed that the Nlgn2-Nlgn1 Y770A mutant displayed a greater degree of co-localization with Gephyrin than the Nlgn2-Nlgn1DelGeph mutant (Fig. EV2E), and expression levels no significant change (Fig. EV2F). These data show that the Gephyrin-binding sequence is critical for proper Nlgn2 function in inhibitory synapses but that either the interaction of Nlgn2 with Gephyrin alone may not be functionally necessary or that the

Gephyrin interaction mediated by Y770 phosphorylation may be less relevant in a cellular context than in vitro.

## An intracellular sequence adjacent to the Gephyrin-binding region is necessary for Nlgn2 inhibitory function

Given the relative brevity of the intracellular Nlgn sequence, we set out to map the function of other intracellular Nlgn2 regions more comprehensively. To this end, we constructed a series of expanding Nlgn2 deletion constructs: Nlgn2-mt1 (Nlgn2 PDZ domain with the Nrxn PDZ domain) to test whether PSD95 and S-SCAM interaction with the Nlgn2 PDZ domain affects Nlgn2 inhibitory function; Nlgn2-mt2 (delete proline-rich domain) to test whether collybistin binding with the Nlgn2 proline-rich domain affects Nlgn2 inhibitory function; Nlgn-mt3 (delete a 21-residue before the gephyrin binding domain); Nlgn-mt4 and Nlgn5 mt are deleted large part of Nlgn2 except gephyrin binding domain (Fig. 4A,B). All constructs were confirmed to be localized to the neuronal cell surface (Fig. EV3A). First, we assessed the specificity of the PDZ domain using the Nlgn2-mt1 construct and collybistin domain using the Nlgn2-mt2 construct and found that this construct fully rescued the IPSC phenotype (Fig. 4C,D). We then tested Nlgn2-mt5 and Nlgn2-mt4, which proved to abolish the Nlgn2 function (Fig. 4C,D). Subsequent smaller deletions demonstrated that a 21-residue was necessary for the Nlgn2 function in addition to the previously identified Gephyrin interacting domain (Fig. 4C,D). No interactor is known for this sequence, suggesting that other intracellular mechanisms in addition to gephyrin binding enable the inhibitory synapse function of Nlgn2.

## The extracellular domain of Nlgn2 is required for inhibitory synapse function

So far, we could establish that the intracellular domain of Nlgn2 is necessary for its function in GABAergic synaptic transmission but

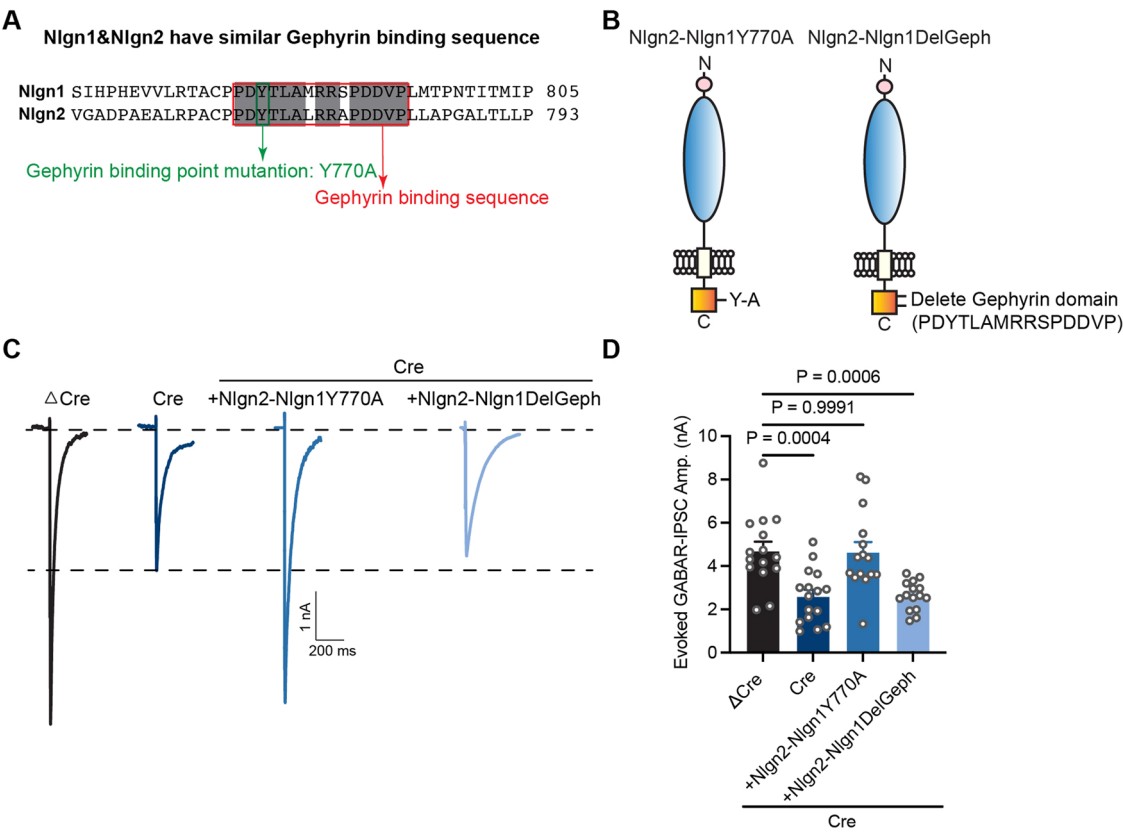

**Figure 3. The cytoplasmic gephyrin binding motif is required for Nlgn2 function whereas tyrosine phosphorylation is not.**

(A) Alignment of Nlgn1 and Nlgn2 gephyrin binding sequence. (B) Schematic of Nlgn2-Nlgn1Y770A (Nlgn2 extracellular domain with Nlgn1 intercellular domain with a gephyrin binding point mutation), Nlgn2-Nlgn1DelGeph (Nlgn2 extracellular domain with Nlgn1 intercellular domain truncation of the entire gephyrin binding sequence) constructs. (C) Representative traces of evoked GABAR-IPSC recorded from DIV14-16 cultured Nlgn1234 conditional knockout mice neurons in the four conditions (ΔCre, Cre, Cre+Nlgn2-Nlgn1Y770A, Cre+Nlgn2-Nlgn1DelGeph). (D) Summary graph of evoked GABAR-IPSC amplitude in all conditions (Bar and line graphs indicate mean ± SEM; numbers of cells/experiments = 15/3,16/3,15/3, and 15/3 for each column, left to right). Nonsignificant $p > 0.05$; ***$p < 0.001$, one-way ANOVA with post hoc Dunnett's Multiple comparisons. Nonsignificant relations are indicated as ns. Source data are available online for this figure.

that, at the same time, the Nlgn1 intracellular domain functionally substituted for the intracellular domain of Nlgn2. Thus, the Nlgn2 intracellular domain does not determine its inhibitory synapse specificity. Therefore, we asked whether Nlgn2's intracellular domain would be sufficient to recruit Nlgn1 to function in GABAergic neurotransmission and created a Nlgn1-Nlgn2 construct consisting of extracellular Nlgn1 and intracellular Nlgn2 domains (Fig. 5A). Electrophysiological recordings showed that the Nlgn1-Nlgn2 chimeric construct was unable to rescue the IPSC phenotype in Nlgn1-4 cKO neurons (Fig. 5B,C). Again, we confirmed that Nlgn1-Nlgn2 properly localizes to the cell surface (Fig. EV2A). This observation suggests that the extracellular domain of Nlgn2 plays a crucial role in determining the specificity of Nlgn2 function in GABAergic synaptic transmission.

## Nlgn2 binding to MDGAs does not confer inhibitory synapse specificity to Nlgn2

To examine what extracellular sequences mediate the inhibitory synapse function of Nlgn2, we constructed a series of Nlgn2 variants with informative mutations in the extracellular domain. We introduced into Nlgn2 a series of 5 point mutations (Q370A/

E372A/L374A/N375A/D377A) that are predicted to abolish Nrxn binding to Nlgn2 based on previous results for Nlgn1 (Ko et al, 2009), and a second set of 3 point mutations (F433A/M434A/W438A) that correspond to Nlgn1 mutations which block dimerization (Ko et al, 2009). Unfortunately, these two sets of Nlgn2 mutations prevented surface transport of Nlgn2, suggesting that they may hinder proper folding of Nlgn2, which limited their usefulness (Fig. EV3B,C).

Recently the cell-adhesion molecules MDGA1 and MDGA2 were shown to compete with Nrxns for binding to Nlgns (Connor et al, 2016; Gangwar et al, 2017; Kim et al, 2017; Lee et al, 2013). Crystal structures of the MDGA1/Nlgn2 complex provided critical insights into this interaction, showing that the two N-terminal Ig domains of MDGA1 bind to the Nlgn2 homodimer (Elegheert et al, 2017b; Gangwar et al, 2017; Kim et al, 2017). To interrogate whether the interaction of MDGAs with the extracellular domain of Nlgn2 confers synaptic specificity, we generated a Nlgn2 mutant (Nlgn2-MDGA1mt) containing three point mutations H278A, D362K, E372K that were previously shown to interfere with MDGA1 binding (Fig. 5D) (Gangwar et al, 2017). The Nlgn2-MDGA1mt construct properly localized to the neuronal surface and co-localized with Gephyrin puncta (Fig. EV2A,B). In line with

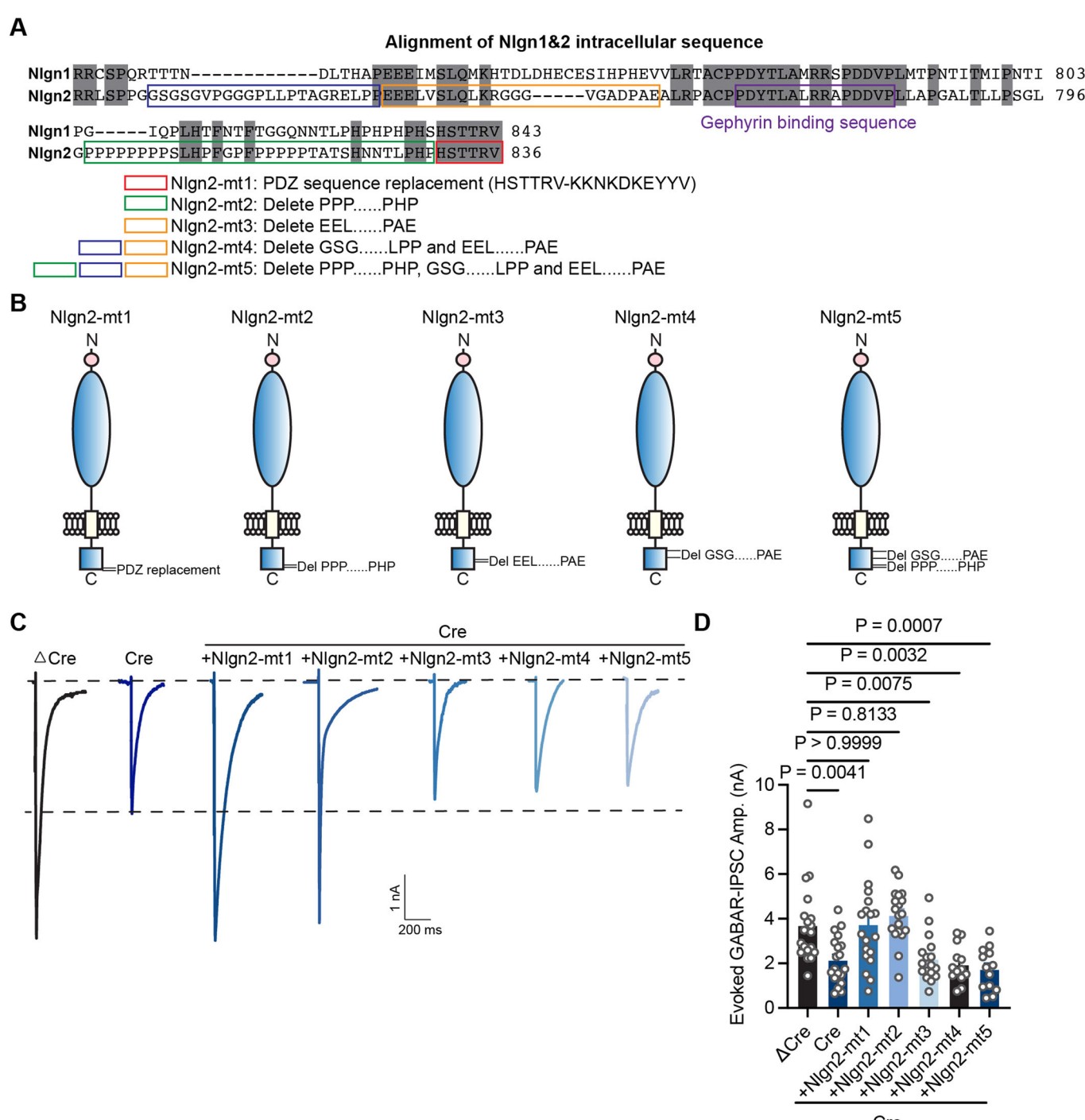

Figure 4. A conserved cytoplasmic 21 residue sequence is necessary for Nlgn2 function, whereas the PDZ-domain binding sequence is dispensable.

(A) Alignment of Nlgn1 and Nlgn2 intracellular domain. Colored boxes show the different Nlgn2 intracellular domain truncations. (B) Schematic of Nlgn2-mt1, Nlgn2-mt2, Nlgn2-mt3, Nlgn2-mt3, Nlgn2-mt4 and Nlgn2-mt5 constructs. (C) Representative traces of evoked GABAR-IPSC recorded from DIV14-16 cultured Nlgn1234 conditional knockout mice neurons in the seven conditions (ΔCre, Cre, Cre+Nlgn2-mt1, Cre+Nlgn2-mt2, Cre+Nlgn2-mt3, Cre+Nlgn2-mt4, Cre+Nlgn2-mt5). (D) Summary graph of evoked GABAR-IPSC amplitude in all conditions (Bar and line graphs indicate mean ± SEM; numbers of cells/experiments = 19/4,19/4,19/4,19/4,17/4,13/3, and 13/3 for each column, left to right). Nonsignificant $p > 0.05$; **$p < 0.01$; ***$p < 0.001$, one-way ANOVA with post hoc Dunnett's Multiple comparisons. Nonsignificant relations are indicated as ns. Source data are available online for this figure.

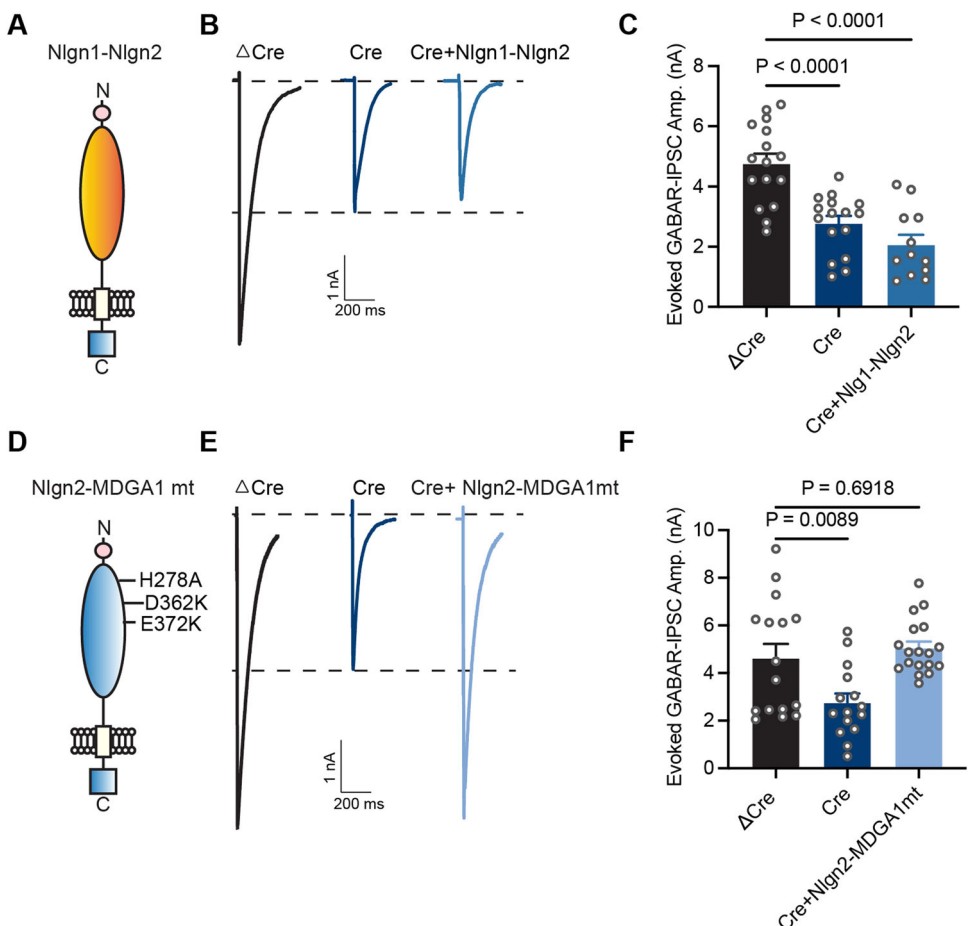

**Figure 5. Different from the functional promiscuity of the Nlgn1 and Nlgn2 cytoplasmic sequences, their extracellular sequences encode synapse type-specificity that is independent of MDGA binding.**

(A) Schematic of Nlgn1-Nlgn2 (Nlgn1 extracellular domain with Nlgn2 intercellular domain) constructs. (B) Representative traces of evoked GABAR-IPSC recorded from DIV14-16 cultured Nlgn1234 conditional knockout mice neurons in the three conditions (ΔCre, Cre, Cre+Nlgn1-Nlgn2). Note that we rescued Nlgn1-Nlgn2 here from the identical three batches of Fig. 2, so the ΔCre and Cre traces here are the same as in Fig. 2. (C) Summary graph of evoked GABAR-IPSC amplitude in all conditions (Bar and line graphs indicate mean ± SEM; numbers of cells/experiments = 16/3,16/3, and 12/3 for each column, left to right). ****$p < 0.0001$, one-way ANOVA with post hoc Dunnett's Multiple comparisons. Nonsignificant relations are indicated as ns. Note that we rescued Nlgn1-Nlgn2 here from the identical three batches of Fig. 2, so the ΔCre and Cre data here are the same as in Fig. 2. (D) Schematic of Nlgn2-MDGA1mt (Nlgn2 with MDGA1-binding point mutation) constructs. (E) Representative traces of evoked GABAR-IPSC recorded from DIV14-16 cultured Nlgn1234 conditional knockout mice neurons in the three conditions (ΔCre, Cre, Cre+Nlgn2-MDGA1mt). (F) Summary graph of evoked GABAR-IPSC amplitude in all conditions (Bar and line graphs indicate mean ± SEM; numbers of cells/experiments = 16/3,15/3, and 18/3 for each column, left to right). Nonsignificant $p > 0.05$; **$p < 0.01$, one-way ANOVA with post hoc Dunnett's Multiple comparisons. Nonsignificant relations are indicated as ns. Source data are available online for this figure.

these findings, functional rescue experiments revealed that the Nlgn2-MDGA1mt construct fully retained the ability of Nlgn2 to rescue the IPSC impairment of Nlgn1234 cKO neurons (Fig. 5E,F). These data suggest that MDGA1 does not directly suppress the inhibitory function of Nlgn2 in synapses. This finding is consistent with recent work showing that MDGA1 targets APP, but not Nlgn2, in hippocampal CA1 GABAergic neural circuits (Kim et al, 2022).

## The extracellular domain of Nlgn1 is sufficient for the excitatory synapse specificity and function of Nlgn1

Next, we asked whether the domain requirements we observed for Nlgn2 also apply to other Nlgns. Therefore, we turned to Nlgn1

that is known to function exclusively in excitatory synapses (Chanda et al, 2017; Chubykin et al, 2007; Gan and Sudhof, 2020; Ko et al, 2009; Song et al, 1999) using equivalent experiments with Nlgn1234 cKO hippocampal neurons expressing Cre-EGFP. As expected, we observed a significant decrease in both AMPAR- and NMDAR-mediated excitatory synaptic responses when we compared Cre-infected with ΔCre-infected neurons (Fig. 6B–D) (Chanda et al, 2017; Gan and Sudhof, 2020). The phenotype was more pronounced in NMDAR responses (~63% decrease) compared to AMPAR responses (~39% decrease) (Fig. 6B,C), which is consistent with previous findings (Chanda et al, 2017; Gan and Sudhof, 2020).

To investigate Nlgn1 function, we performed rescue expression experiment with various Nlgn1 variants (Fig. 6A). As expected, the

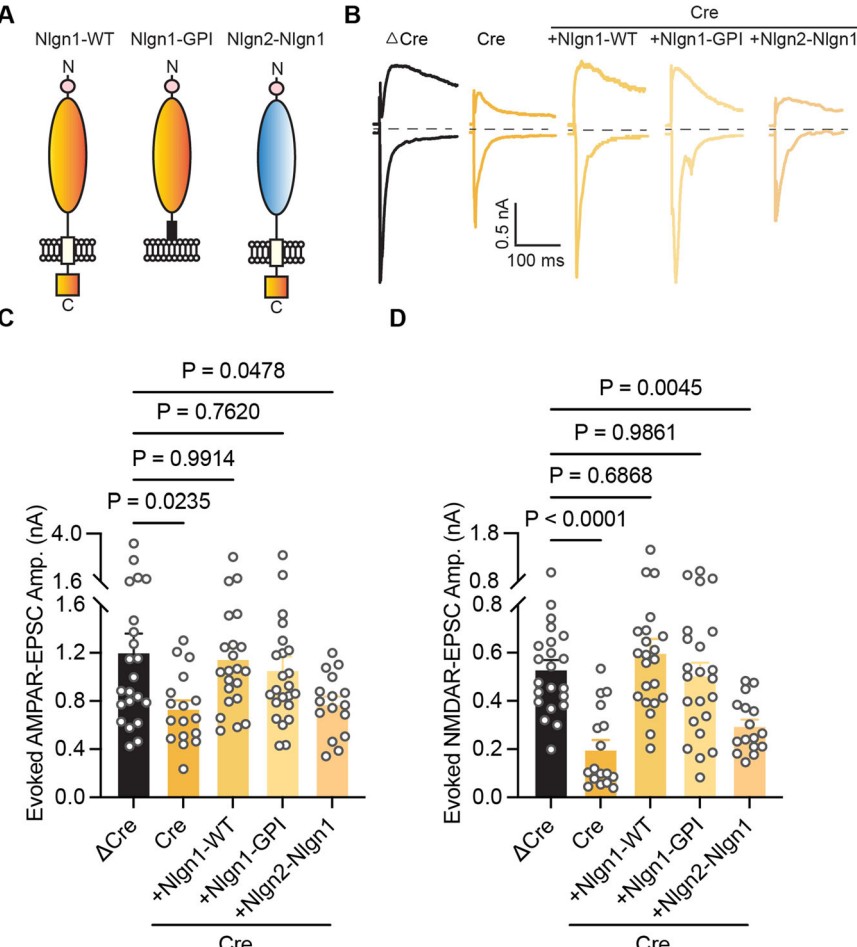

**Figure 6. Nlgn1 extracellular domain provides the specificity of Nlgn1 excitatory function.**

(A) Schematic of Nlgn1-WT, Nlgn1-GPI (Nlgn1 chimeric construct only has extracellular domain), Nlgn2-Nlgn1 constructs. (B) Representative traces of evoked AMPAR- and NMDAR-EPSCs recorded from DIV14-16 cultured Nlgn1234 conditional knockout mice neurons. AMPAR-EPSCs and NMDAR-EPSC recorded at −70 mV and +40 mV, respectively, in the five conditions (ΔCre, Cre, Cre+Nlgn1-WT, Cre+Nlgn1-GPI, Cre+Nlgn2-Nlgn1). (C) Summary graph of evoked AMPAR-EPSC amplitude in all conditions (Bar and line graphs indicate mean ± SEM; numbers of cells/experiments = 22/4, 17/4, 22/4, 23/4, and 16/3 for each column, left to right). Nonsignificant $p > 0.05$; *$p < 0.05$, one-way ANOVA with post hoc Dunnett's Multiple comparisons. Nonsignificant relations are indicated as ns. (D). Summary graph of evoked NMDAR-EPSC amplitude in all conditions (Bar and line graphs indicate mean ± SEM; numbers of cells/experiments = 22/4, 17/4, 22/4, 23/4, and 16/3 for each column, left to right). Nonsignificant $p > 0.05$; **$p < 0.01$; ****$p < 0.0001$, one-way ANOVA with post hoc Dunnett's Multiple comparisons. Nonsignificant relations are indicated as ns. Source data are available online for this figure.

excitatory synaptic phenotype could be fully restored by full-length Nlgn1 expression (Fig. 6B–D). However, in contrast to what we had observed for Nlgn2, the Nlgn1-GPI construct (Nlgn1 without an intracellular domain) also rescued the excitatory synapse phenotype. Consistent with a lack of importance of the intracellular Nlgn1 domain, a Nlgn2-Nlgn1 construct was unable to rescue the glutamatergic synaptic phenotype (Fig. 6B–D). Importantly, we confirmed the surface expression Nlgn1-GPI and Nlgn2-Nlgn1. Co-localization of the constructs with Homer1 revealed that a larger fraction of Nlgn1-GPI, and a smaller fraction of Nlgn2-Nlgn1 were co-localized with Homer1 (Fig. EV4A,B), and expression levels no significant change (Fig. EV4C). We also found Nlgn1-Nlgn2 is sufficient for the glutamatergic synaptic transmission function of Nlgn1, but Nlgn2-WT doesn't (Fig. EV5). These findings demonstrate that—unlike for Nlgn2—the extracellular domain of

Nlgn1 is sufficient for Nlgn1 function and also confers specificity to excitatory synaptic transmission.

## Nrxn- and MDGA1-binding and dimerization of Nlgn1 are not required for glutamatergic synaptic transmission

The observation that the extracellular domain of Nlgn1 is critical for its excitatory synapse function suggests that its physical interaction with pre-synaptic binding partners is important. Known Nlgn1-binding partners include Nrxns and MDGAs. To test the necessity of these interactions, we generated Nlgn1 variants that are unable to bind Nrxns and MDGA1, respectively (Arac et al, 2007; Ko et al, 2009). To more generally assess the importance of extracellular Nlgn1 interactions, we also generated a Nlgn1 variant that is unable to dimerize (Fig. 7A). All constructs were

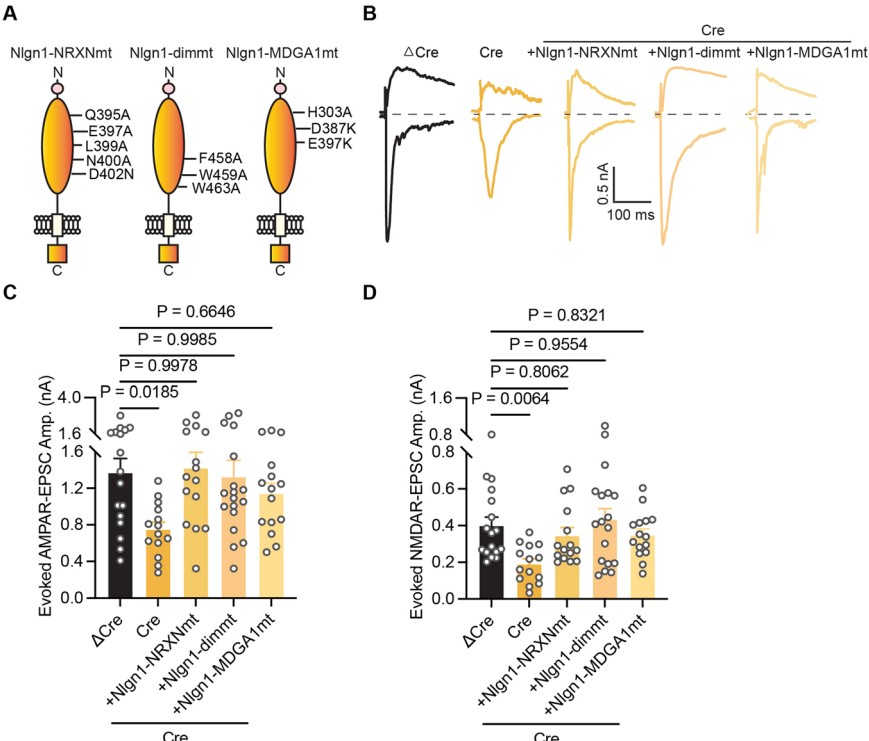

**Figure 7. Nlgn1 function in excitatory synapses does not require Neurexin-binding, MDGA-binding, or dimerization.**

(A) Schematic of Nlgn1-NRXNmt (Nlgn1 with neurexin-binding point mutation), Nlgn1-dimmt (Nlgn1 dimerization point mutation), Nlgn1-MDGA1mt (Nlgn1 with MDGA1-binding point mutation) constructs. (B) Representative traces of evoked AMPAR- and NMDAR-EPSCs recorded from DIV14-16 cultured Nlgn1234 conditional knockout mice neurons. AMPAR-EPSCs and NMDAR-EPSC recorded at −70 mV and +40 mV, respectively, in the five conditions (ΔCre, Cre, Cre+Nlgn1-NRXNmt, Cre+Nlgn1-dimmt, Cre+Nlgn1-MDGA1mt). (C) Summary graph of evoked AMPAR-EPSC amplitude in all conditions (Bar and line graphs indicate mean ± SEM; numbers of cells/experiments = 17/4, 14/4, 15/4, 18/4, and 15/4 for each column, left to right). Nonsignificant $p > 0.05$; *$p < 0.05$, one-way ANOVA with post hoc Dunnett's Multiple comparisons. Nonsignificant relations are indicated as ns. (D) Summary graph of evoked NMDAR-EPSC amplitude in all conditions (Bar and line graphs indicate mean ± SEM; numbers of cells/experiments = 17/4, 14/4, 15/4, 18/4, and 15/4 for each column, left to right). Nonsignificant $p > 0.05$; **$p < 0.01$, one-way ANOVA with post hoc Dunnett's Multiple comparisons. Nonsignificant relations are indicated as ns. Source data are available online for this figure.

appropriately expressed on the neuronal surface (Fig. EV4D). Also, and remarkably, all constructs exhibited a high degree of co-localization with Homer1 (Fig. EV4E), and expression levels no significant change (Fig. EV4F). In concordance with these results, electrophysiological measurements demonstrated that all three constructs fully rescued the excitatory phenotype of Nlgn1234 cKO neurons (Fig. 7B–D). These results suggest the surprising notion that neither Nrxn- nor MDGA1-binding and not even dimerization of Nlgn1 is necessary for Nlgn1 function despite the functional necessity of its extracellular, but not its intracellular domain.

## Discussion

Nlgn1 and Nlgn2 function selectively in glutamatergic excitatory and GABAergic inhibitory synapses, respectively, but how their synapse specificity is determined remains unclear. In this study, we sought to investigate the mechanisms that confer this intriguing synapse specificity onto Nlgn1 and Nlgn2 and to understand how two closely related proteins can possess such contrasting functions. To eliminate the potentially confounding effects of other Nlgns that may heterodimerize with exogenously expressed Nlgns or otherwise influence a specific Nlgn under investigation, we here resorted to a

reductionistic approach that employs a primary neuronal culture system lacking endogenous Nlgns. Specifically, we used as a general substrate of functional analyses neurons that lack all four Nlgns, in which we then expressed a specific Nlgn mutant. Surprisingly, we found that different molecular mechanisms guide Nlgn1 and Nlgn2 function in this system. Our results allow us to draw five main conclusions.

First, deletion of all four Nlgns results in a major dysfunction of both GABAergic and glutamatergic synapses, demonstrating that neuronal Nlgns are crucial for proper synaptic transmission. These results are consistent with previous findings using Nlgn123 cKO and Nlgn1234 cKO mice (Chanda et al, 2017; Gan and Sudhof, 2020). Our findings further corroborate the notion that Nlgn1 is important for glutamatergic and Nlgn2 for GABAergic neuro-transmission since Nlgn1 could selectively rescue the glutamatergic defect and Nlgn2 the GABAergic defect of Nlgn1234 cKO neurons. Moreover, Nlgn1 molecules were primarily targeted to Homer1-positive excitatory synapses and Nlgn2 to Gephyrin-positive inhibitory synapses.

Second, Nlgn2's function at inhibitory synapses requires its intracellular domain. Given this finding, interactions mediated by the intracellular domain would represent an attractive mechanism to target Nlgn2 to GABAergic synapses given the many inhibitory

synapse-specific scaffolding proteins that are known to bind to the Nlgn2 cytoplasmatic domain (Poulopoulos et al, 2009; Tyagarajan et al, 2011). Unexpectedly, however, the Nlgn2 intracellular domain was unable to confer onto Nlgn2 its GABAergic synapse specificity since the Nlgn1 intracellular domain was fully capable of substituting for the Nlgn2 intracellular domain. A conspicuous similarity between Nlgn1 and Nlgn2 is the Gephyrin-binding sequence that is equally present in the Nlgn1 and Nlgn2 cytoplasmic domain despite the fact that Nlgn1 exclusively functions in excitatory synapses lacking Gephyrin. Gephyrin is a key postsynaptic scaffolding protein at GABAergic synapses that enables the recruitment of $GABA_A$ receptors to synapses (Choii and Ko, 2015; Tyagarajan and Fritschy, 2014). Importantly, we found that the Gephyrin-interacting domain is necessary for Nlgn2 function, suggesting that recruitment by Gephyrin is indeed important. However, Gephyrin may not be the only important Nlgn2 binding partner. A Nlgn2 variant that carries the Y770A mutation, previously shown to disrupt Gephyrin binding (Giannone et al, 2013; Poulopoulos et al, 2009), was still fully functional. This finding suggests that potentially other co-factors may stabilize the Gephyrin-Nlgn2 interaction overcoming the Y770A mutation. A previous study also showed an only partial functional impairment of Nlgn2 by the Y770A mutation (Nguyen et al, 2016). However, in this paper the deletion of the entire Gephyrin binding sequence also only partly impaired Nlgn2 function whereas in our experiments such a deletion was functional deleterious. Overall, these results prompted us to look for additional cytosolic Nlgn2 sequences that are important for Nlgn2 function. Through targeted mutagenesis, we identified a new sequence of 21-residue that is just as critical as the better characterized Gephyrin-binding sequence and may represent a binding surface for additional postsynaptic proteins at GABAergic synapses. Of note, the two amino acid residues K749 and R750 were shown to be ubiquitinated and methylated, respectively (Guo et al, 2014; Wagner et al, 2012).

Third, we found that different Nlgns use distinct functional mechanisms at their target synapses. Whereas Nlgn2 requires its intracellular domain for enabling inhibitory synaptic connections, Nlgn1—consistent with previous results (Jiang et al, 2017; Wu et al, 2019)—does not. This may be our conceptually most intriguing result: Despite the fact that Nlgn1 and Nlgn2 are both enabling synapse function, albeit at different types of synapses, their mechanisms of action differ critically.

Fourth, the extracellular but not the intracellular domain of Nlgn1 and Nlgn2 determines their specificity for glutamatergic and GABAergic synapses, respectively. The intracellular domain of Nlgn2 (but intriguingly not of Nlgn1) is necessary for synaptic localization and its "specific" recruitment to synapses. Chimeric Nlgn2-Nlgn1 molecules enabled GABAergic synaptic transmission, but not glutamatergic, while chimeric Nlgn1-Nlgn2 molecules mediated glutamatergic, but not GABAergic synaptic transmission. The specificity for glutamatergic vs. GABAergic synapses is dictated by the Nlgn1, and Nlgn2 extracellular domains, respectively. Nrxns are well-studied pre-synaptic binding partners of Nlgns and thus could provide synapse specificity. However, we found that Nlgn1 variants unable to bind Nrxns still rescued the excitatory phenotypes as reported previously (Ichtchenko et al, 1995; Ichtchenko et al, 1996; Jiang et al, 2017; Ko et al, 2009; Sudhof, 2017; Wu et al, 2019). These results suggest that Nlgns function via interactions with other binding partners in addition to Nrxns. Indeed, recently MDGA proteins have been identified as Nlgn interaction partners, binding through their lg1-lg2 domains to the lobes of the Nlgn extracellular dimer at sites overlapping with the Nrxn binding interface (Elegheert et al, 2017a; Gangwar et al, 2017; Kim et al, 2017). They thus emerge as candidates that may provide synapse specificity for Nlgns. Remarkably, however, we found that mutant Nlgn1 predicted to lack MDGA binding was fully able to rescue Nlgn1 function, indicating that this class of interaction is also not providing specificity. In addition, in vivo experiments show that MDGA1 targets APP, but not Nlgn2, in hippocampal CA1 GABAergic neural circuits to regulate GABAergic synaptic transmission (Kim et al, 2022). In addition to MDGAs, at least Nlgn3 has been shown to bind to PTPRD (Yoshida et al, 2021), which in turn binds to neurexins (Han et al, 2018), suggesting a possible synapse maintenance mechanism involving multiple trans-synaptic interactions.

Finally, we conclude that Nlgn1 may be acting as a monomer in regulating synaptic transmission and that no binding event that involves signal transduction mediated by dimerization plays a role in Nlgn1's specificity. Mutations disrupting Nlgn1 dimerization did not impair Nlgn1 function in our hippocampal cell assay. Intriguingly, a previous study showed that Nlgn1 overexpression enhances both NMDAR- and AMPAR- EPSCs in neurons, even when dimerization is disrupted (Ko et al, 2009).

Our studies also raise new questions. With the conclusion that extracellular mechanisms must be responsible for functional specificity of Nlgns, but neither Nrxn nor MDGA interactions are themselves sufficient to account for the specificity of Nlgns, there must be other thus-far unidentified proteins present at excitatory and inhibitory pre-synaptic compartments that interact with Nlgns. In the case of Nlgn2, the intracellular domain is critical for synaptic function, but its exact role continues to be unclear. For example, Nlgn2 activation by an extracellular signal could stimulate a defined signal-transduction cascade, akin to a conventional receptor/ligand-induced signaling pathway. Unraveling these signaling mechanisms will contribute to a comprehensive understanding of the molecular pathways governing Nlgn-mediated synaptic processes. These questions are crucial for a general understanding of the diversity of Nlgns functions. Detailed molecular, genetic, and biochemical approaches may lead to a better insight into these questions.

# Methods

**Reagents and tools table**

| Reagent/Resource | Reference or Source | Identifier or Catalog Number |
|---|---|---|
| **Experimental Models** | | |
| Nlgn1234 cKO mouse | PMID: 32973045 PMID: 30871858 | N/A |
| **Recombinant DNA** | | |
| FSW-hSyn-NLS-EGFP-ΔCre | This study | N/A |

| Reagent/Resource | Reference or Source | Identifier or Catalog Number |
| --- | --- | --- |
| FSW-hSyn-NLS-EGFP-Cre | This study | N/A |
| FSW-hSyn-HA-Nlgn1-WT | This study | N/A |
| FSW-hSyn-HA-Nlgn1-GPI | This study | N/A |
| FSW-hSyn-HA-Nlgn1-Nlgn2 | This study | N/A |
| FSW-hSyn-HA-Nlgn1-Nrxnmt | This study | N/A |
| FSW-hSyn-HA-Nlgn1-dimmt | This study | N/A |
| FSW-hSyn-HA-Nlgn1-MDGA1mt | This study | N/A |
| FSW-hSyn-HA-Nlgn2-WT | This study | N/A |
| FSW-hSyn-HA-Nlgn2-GPI | This study | N/A |
| FSW-hSyn-HA-Nlgn2-Nlgn1 | This study | N/A |
| FSW-hSyn-HA-Nlgn2-Nrxnmt | This study | N/A |
| FSW-hSyn-HA-Nlgn2-dimmt | This study | N/A |
| FSW-hSyn-HA-Nlgn2-MDGA1mt | This study | N/A |
| FSW-hSyn-HA-Nlgn2mt1 | This study | N/A |
| FSW-hSyn-HA-Nlgn2mt2 | This study | N/A |
| FSW-hSyn-HA-Nlgn2mt3 | This study | N/A |
| FSW-hSyn-HA-Nlgn2mt4 | This study | N/A |
| FSW-hSyn-HA-Nlgn2mt5 | This study | N/A |
| FSW-hSyn-HA-Nlgn2-Nlgn1 Y-A | This study | N/A |
| FSW-hSyn-HA-Nlgn2-Nlgn1 dele Geph | This study | N/A |
| **Antibodies** | | |
| Mouse anti-HA | BioLegend | Cat. #: 901501 |
| Rabbit anti-HA | Cell Signaling Technologies | Cat. #: 3724S |
| Guinea pig anti-Gephyrin | Synaptic Systems | Cat. #: 147318 |
| Mouse anti-Gephyrin | Synaptic Systems | Cat. #: 147011 |
| Rabbit anti-Homer1 | Millipore | Cat. #: ABN37 |
| Guinea pig anti-Homer1 | Synaptic Systems | Cat. #: 160005 |
| Chicken anti-MAP2 | Encor | Cat. #: CPCA-MAP2 |
| Mouse anti-Nlgn1 | Synaptic Systems | Cat. #: ab308451 |
| Mouse anti-Nlgn2 | Synaptic Systems | Cat. #: ab317510 |
| Rabbit anti-Nlgn3 | PMID: 32973045 | N/A |
| Rabbit anti-Tuj1 | BioLegend | Cat. #: 802001 |
| Goat anti-Guinea Pig IgG (H + L) Secondary Antibody, Alexa Fluor™ 546 | Thermo Fisher | Cat. #: A-11074 |
| Goat anti-Rabbit IgG (H + L) Highly Cross-Adsorbed Secondary Antibody, Alexa Fluor™ 647 | Thermo Fisher | Cat. #: A-21245 |
| Goat anti-Chicken IgY (H + L) Secondary Antibody, Alexa Fluor™ 488 | Thermo Fisher | Cat. #: A-11039 |

| Reagent/Resource | Reference or Source | Identifier or Catalog Number |
| --- | --- | --- |
| Goat anti-Mouse IgG (H + L) Highly Cross-Adsorbed Secondary Antibody, Alexa Fluor™ 647 | Thermo Fisher | Cat. #: A-21236 |
| Goat anti-Rabbit IgG (H + L) Cross-Adsorbed Secondary Antibody, Alexa Fluor™ 546 | Thermo Fisher | Cat. #: A-11010 |
| **Chemicals, Enzymes and other reagents** | | |
| Ara-c | Sigma-Aldrich | Cat. #: C6645 |
| B-27 supplement | Thermo Fisher | Cat. #: 12587010 |
| Matrigel | BD Biosciences | Cat. #: 356230 |
| Neurobasal-A Medium | Thermo Fisher | Cat. #: 10888022 |
| Minimum Essential Medium | Thermo Fisher | Cat. #: 12587010 |
| Glucose | Millipore-Sigma | Cat. #: G7021 |
| Fluoromount-G | SouthernBiotech | Cat. #: 00-4958-02 |
| QX-314 | Tocris | Cat. # 1014 |
| APV | Millipore Sigma | Cat. # A8054 |
| CNQX | Millipore Sigma | Cat. # C127 |
| Picrotoxin | Millipore Sigma | Cat. # 1675 |
| **Software** | | |
| ImageStudio software | Li-Cor | RRID: SCR_013715 |
| Image J | NIH | RRID: SCR_003070 |
| Nikon NIS-Elements | Nikon | N/A |
| GraphPad Prism | GraphPad Software Inc | N/A |
| Pclamp 10 | Molecular Devices | RRID: SCR_011323 |
| Adobe illustrator | Adobe | N/A |

## Mouse breeding and husbandry

All animal experiments were performed with male and female newborn mice according to institutional guidelines and approved by the Administrative Panel on Laboratory Animal Care of Stanford University School of Medicine (Protocol ID: 18846). A detailed description of generating Nlgn1234 cKO mice is described in the previous study (Wu et al, 2019). All experiments except for Fig. EV1, were performed in a "blinded" fashion (i.e., the experimenter was unaware of whether a sample represented a test or control sample).

## Plasmids and lentivirus preparation

All plasmids were in a pFSW67 backbone and used the human synapsin promoter, for all neuroligin added an N-terminal HA

peptide fused to the mature coding sequence. Lentivirus were prepared as described previously (Chanda et al, 2017). Briefly, the lentiviral expressing vectors (12 µg) were co-transfected with three helper plasmids (pRSV-REV (4 µg), pMDLg/pRRE (8 µg), and VSV-G (6 µg)) into HEK293T cells cultured in T-75 flasks using calcium phosphate transfection. After a complete medium exchange at 7 h, supernatants were subsequently collected at 48 h, filtered via 0.45 µm filter, and ultracentrifuged at $64{,}000 \times g$ for 2 h. Pellets were resuspended in MEM and stored at −80 °C before use. All viruses used in this paper are as below:

Lenti-Syn-ΔCre-EGFP

Lenti-Syn-Cre-EGFP

Lenti-Syn-HA-Nlgn1-WT (Nlgn1-wild-type)

Lenti-Syn-HA-Nlgn1-GPI (We deleted all transmembrane regions and cytoplasmic sequences of Nlgn1 and attaches its extracellular domains to the membrane using a GPI-anchor.)

Lenti-Syn-HA-Nlgn1-Nlgn2 (We made the extracellular domain of Nlgn1 transplanted onto Nlgn2 intercellular.)

Lenti-Syn-HA-Nlgn1-Nrxnmt (Nlgn1 with Nrxn binding site mutant was generated by making previously described mutations Q395A/E397A/L399A/N400A/D402A (Arac et al, 2007).)

Lenti-Syn-HA-Nlgn1-dimmt (Nlgn1 dimerization mutant was generated by making previously described mutations F458A/M459A/W463A (Arac et al, 2007; Ko et al, 2009).)

Lenti-Syn-HA-Nlgn1-MDGA1mt (We aligned the Nlgn1 and Nlgn2 sequences, made the Nlgn1 with MDGA1 binding site mutations H303A/D387K/E397K.)

Lenti-Syn-HA-Nlgn2-WT (Nlgn2-wild-type)

Lenti-Syn-HA-Nlgn2-GPI (We deleted all transmembrane regions and cytoplasmic sequences of Nlgn2 and attaches its extracellular domains to the membrane using a GPI-anchor.)

Lenti-Syn-HA-Nlgn2-Nlgn1 (We made extracellular domain of Nlgn2 transplanted onto Nlgn1 intercellular.)

Lenti-Syn-HA-Nlgn2-Nrxnmt (We aligned the Nlgn1 and Nlgn2 sequences, made the Nlgn2 with Nrxn binding site mutations Q370A/E372A/L374A/N375A/D377A.)

Lenti-Syn-HA-Nlgn2-dimmt (We aligned the Nlgn1 and Nlgn2 sequences, made the Nlgn2 dimerization mutations F433A/M434A/W438A.)

Lenti-Syn-HA-Nlgn2-MDGA1mt (Nlgn2 with MDGA1 binding site mutant was generated by making previously described mutations H278A/D362K/E372K (Gangwar et al, 2017).)

Lenti-Syn-HA-Nlgn2mt1 (We replaced the Nlgn2-PDZ domain HSTTRV with Nrxn-PDZ domain KKNKDKEYYV.)

Lenti-Syn-HA-Nlgn2mt2 (We deleted Nlgn2 intracellular sequences PPPPPPPSLHPFPPPPPTATSHNNTLPHP.)

Lenti-Syn-HA-Nlgn2mt3 (We deleted Nlgn2 intracellular sequences EEELVSLQLKRGGGVGADPAE.)

Lenti-Syn-HA-Nlgn2mt4 (We deleted Nlgn2 intracellular sequences GSGSGVPGGGPLLPTAGRELPEEEELVSLQLKRGGGVGADPAE.)

Lenti-Syn-HA-Nlgn2mt5 (We deleted Nlgn2 intracellular sequences GSGSGVPGGGPLLPTAGRELPEEEELVSLQLKRGGGVGADPAE and PPPPPPPPSLHPFPPPPPTATSHNNTLPHP.)

Lenti-Syn-HA-Nlgn2-Nlgn1 Y-A (We made extracellular domain of Nlgn2 transplanted onto Nlgn1 intercellular and made gephyrin binding point mutation as previously described Y770A (Poulopoulos et al, 2009).)

Lenti-Syn-HA-Nlgn2-Nlgn1 dele Geph (We deleted the whole Nlgn1 with gephyrin binding domain sequences PDYTLALR RAPDDVP.).

## Neuronal culture and lentivirus infection

Cell culture of hippocampal primary neurons from both male and female Nlgn1234 cKO newborn (P0) mice was performed as described previously (Wang et al, 2022). Briefly, dissected hippocampus was digested at 37 °C for 25 min with 10U/ml papain in HBSS buffer, washed three times with plating medium (MEM supplemented with 0.5% glucose, 0.02% NaHCO$_3$, 0.1 mg/ml transferrin, 10% FBS, 2 mM L-glutamine, and 0.025% mg/ml insulin), and then gently dissociated in plating medium, and filter with 70 µm cell strainer, and seeded on Matrigel pre-coated coverslip placed inside 24-well plate. The day of plating was considered as 0 days in vitro (DIV 0). After 24 h (DIV 1), 90% of the plating medium was replaced with neuronal growth medium (Neurobasal supplemented with 0.5% glucose, 0.02% NaHCO$_3$, 0.1 mg/ml transferrin, 5% FBS, 2% B27 supplement, and 0.5 mM L-glutamine), At DIV3, 50% of the medium was replaced with fresh growth medium additionally supplemented with finally concertation 4 µM Ara-C. Cultured hippocampal neurons were infected at DIV 4 with lentiviruses expressing ΔCre-EGFP or Cre-EGFP and subsequently were infected at DIV 6 with lentiviruses expressing different neuroligin constructs.

## Immunohistochemistry

All solutions were made fresh and filtered via a 0.22 µm filter prior to staring experiments. For HA surface-labeling experiments were performed as described previously (Trotter et al, 2019). Briefly, primary neurons were first washed at room temperature once with a normal Tyrode's bath solution, and then incubated for 20 min with purified Mouse anti-HA monoclonal antibody (1:500; BioLegend, Cat. #: 901501) or Rabbit anti-HA monoclonal antibody (1:500; Cell Signaling Technologies, Cat. #: 3724S) diluted in normal Tyrode's bath solution. Cultures were then gently washed three times with normal Tyrode's bath solution, followed by fixation for 20 min at 4 °C with 4% PFA. Following fixation, cultures were washed three times with Dulbecco's PBS (DPBS). For surface labeling experiments (All the Nlgn constructs were all incubated live with an anti-HA monoclonal antibody and then fixed and further processed) to be used for conventional imaging, cultures were blocked for 1 h at room temperature with antibody dilution buffer (ADB) without Triton X-100, which contains 5% normal goat serum diluted in DPBS. Cultures were then labeled with Alexa Fluor-conjugated secondary antibody (1:1000; Invitrogen) diluted in ADB for 1 h at room temperature. Next for non-surface labeling, cells were washed and permeabilized and blocked for 1 h with ADB with Triton X-100, which contains 0.2% Triton X-100 and 5% normal goat serum diluted in DPBS. Non-surface primary antibodies Guinea pig anti-Gephyrin monoclonal antibody (1:200, Synaptic Systems, Cat. #: 147318) or Mouse anti-Gephyrin monoclonal antibody (1:1000, Synaptic Systems, Cat. #: 147011)/ Rabbit anti-Homer1 monoclonal antibody (1:500, Millipore, Cat. #: ABN37) or Guinea pig anti-Homer1 monoclonal antibody (1:500, Synaptic Systems, Cat. #: 160005)/Chicken anti-MAP2 monoclonal

antibody (1:1000, Encor, Cat. #: CPCA-MAP2) were diluted in ADB, and cells were incubated overbought at 4 °C or 2 h at room temperature. Cultures were washed three times and then incubated with either Alexa Fluor-conjugated secondary antibodies (1:1000; Invitrogen) in ABD for 1 h. Following three washed, coverslip for conventional imaging were inverted onto glass microscope slides with Fluoromount-G Mounting medium (Southern Biotech). In certain experiments, the number of cells analyzed was relatively low, as indicated in Fig. 1D,F. Despite this, the observed effect size and the consistency of the results suggest that the sample size was sufficient.

## Immunoblotting

Hippocampal neurons in 24-well plates were lysed in RIPA buffer containing 150 mM NaCl, 1% Triton X-100, 0.1% SDS, 25 mM Tris-HCl, pH 7.4, and Complete EDTA-Free Protease Inhibitor Cocktail (Sigma Millipore). Lysates were incubated on ice for 30 min and clarified by centrifugation at $14,000 \times g$ for 30 min at 4 °C. Lysates were boiled in sample buffer containing 1% β-mercaptoethanol at 42 °C for 30 min. Proteins were analyzed by SDS-PAGE using 4–20% Mini-Protean TGX precast gels (Bio-Rad). Proteins were transferred onto nitrocellulose membranes for 7 min at 25 V using the Trans-Blot Turbo transfer system (Bio-Rad). Membranes were blocked in 5% milk diluted in TBST for 1 h at room temperature. Membranes were then incubated overnight at 4 °C with the following primary antibodies diluted in 5% BSA diluted in TBST solution: Mouse anti-Nlgn1 (1:1000, Synaptic Systems, Cat. #: ab308451), Mouse anti-Nlgn2 (1:1000, Synaptic Systems, Cat. #: ab317510), Rabbit anti-Nlgn3 (639B, 1:500, T.C.S. Laboratory), and Rabbit anti-Tuj1 (1:1000, BioLegend, Cat. #: 802001) Membranes were subsequently incubated for 1 h at room temperature with the following compatible secondary antibodies (LI-COR), diluted 1:10,000 in blocking solution: IRDye 800LT donkey anti-mouse or anti-Rabbit; IRDye 800CW donkey anti-mouse or anti-Rabbit. Quantitative analysis was performed by a dual-channel infrared imaging system, an Odyssey Infrared Imager CLX, and Image Studio 5.2.5 software (LI-COR).

## Confocal image acquisition and analysis

Serial confocal z-stack images were acquired using a Nikon confocal microscope (A1RSi) with a 40× objective. Images were analyzed with by NIS-Elements AR analysis software.

## Electrophysiology

Electrophysiological recordings were performed from primary hippocampal culture neurons plated on coverslip, which were placed in a recording chamber mounted on a fixed stage inverted phase-contrast microscope (Olympus). Patch electrodes (3–5 MΩ) were pulled from borosilicate glass capillary tubes (Warner Instruments) using a PC-10 pipette puller (Narishige). Whole-cell capacitance and series resistances were recorded and compensated to >80%, and in addition, series resistances were less than two times the tip resistance. The Tyrode's bath solution contained (in mM): 129 NaCl, 5 KCl, 2 CaCl$_2$, 1 MgCl$_2$, 0.01 glycine, 30 D-glucose and 25 HEPES, pH 7.2–7.4.

All recording under voltage-clamp model with a pipette solution containing (in mM): 135 CsCl, 10 HEPES, 10 EGTA, 2 Mg-ATP, 2 Na$_2$GTP, and 5 QX-314, pH 7.35 (adjusted with CsOH). Presynaptic Action-potential for evoked synaptic responses were triggered by 0.5-ms current (40–90 μA) injections through a local extracellular electrode (FHC concentric bipolar electrode, Catalogue number CBAEC75) placed ~100 μm from the soma of neurons recorded. Inhibitory synaptic currents were made in presence of 20 μM CNQX and 50 μM APV to block AMPAR and NMDAR-mediated currents, respectively. GABAR-IPSCs were recorded at −60 mV and measured at the peak of the current.

Excitatory synaptic currents were made in presence of 100 μM picrotoxin to block inhibitory currents and a small (10 nM) amount of TTX to reduce epileptiform activity. AMPAR-EPSCs were recorded as the peak current (~2 ms window) at −70 mV and NMDAR-EPSCs at +40 mV.

## Quantifications and statistical analyses

Electrophysiological data were analyzed in Clampfit 10.7 (Molecular Devices). For clarity, all stimulus were blanked and not shown in the figures. All data were shown as means ± SEM, numbers of cells and batches analyzed are shown in the bars. Statistical significance (*$p < 0.05$; **$p < 0.01$; ***$p < 0.001$; ****$p < 0.0001$, non-significant comparisons are indicted as NS) was analyzed with Prism 9, GraphPad. Unpaired t test was used for comparison between two groups. One-way ANOVA was used for comparison among more than two groups.

# Data availability

No data have been deposited to public repositories.

The source data of this paper are collected in the following database record: biostudies:S-SCDT-10_1038-S44319-024-00286-4.

# Peer review information

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

## Acknowledgements

This study was supported by grant from the National Institute of Mental Health (NIMH) (2R01MH092931 to MW and TCS).

## Author contributions

**Jinzhao Wang**: Data curation; Formal analysis; Investigation; Methodology; Writing—original draft; Writing—review and editing. **Thomas Sudhof**: Conceptualization; Resources; Supervision; Funding acquisition; Writing—original draft; Writing—review and editing. **Marius Wernig**: Resources; Supervision; Funding acquisition; Writing—original draft; Writing—review and editing.

Source data underlying figure panels in this paper may have individual authorship assigned. Where available, figure panel/source data authorship is listed in the following database record: biostudies:S-SCDT-10_1038-S44319-024-00286-4.

## Disclosure and competing interests statement

The authors declare no competing interests.

# Expanded View Figures

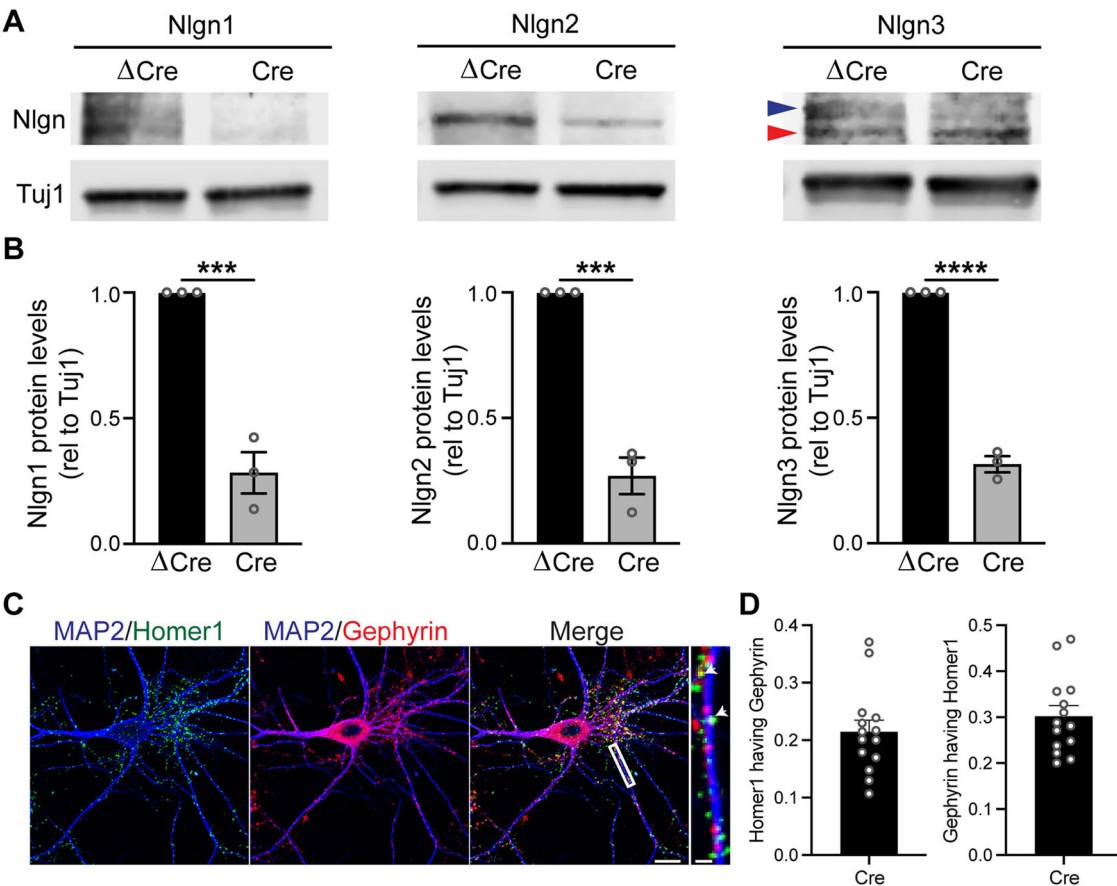

**Figure EV1.** **Neuroligin protein measurement and co-staining Homer1 with Gephyrin in Nlgn1234 cKO cultured hippocampal neurons (related to Fig. 1).**

(A) Representative images of western blot for Nlgn1, Nlgn2, and Nlgn3 protein expressions from DIV14-16 cultured Nlgn1234 conditional knockout mice neurons, infected with either ΔCre or Cre. Note that the Nlgn3 antibody detects Nlgn3 (blue arrow) and nonspecific band (red arrow). (B) Summary graphs of western blot analysis for Nlgn1, Nlgn2, and Nlgn3 protein expressions. (Bar and line graphs indicate mean ± SEM; samples/experiments = 3/3. 3 technical replicates. Statistical significance was assessed by unpaired t test, ***p < 0.001; ****p < 0.0001). (C) Representative image from DIV14-16 cultured Nlgn1234 conditional knockout mice neurons. The neuron was labeled with antibodies to Homer1 (Green), Gephyrin (red), and MAP2 (blue). Scale bar: 20 μm. The right panels show an enlarged box area (arrowheads indicate Homer1 puncta overlapped with Gephyrin puncta) Scale bar: 5 μm. (D) Summary graph of Homer1 overlap percentage with Gephyrin and Gephyrin overlap percentage with Homer1 (Bar and line graphs indicate mean ± SEM; numbers of cells/experiment = 14/1). Source data are available online for this figure.

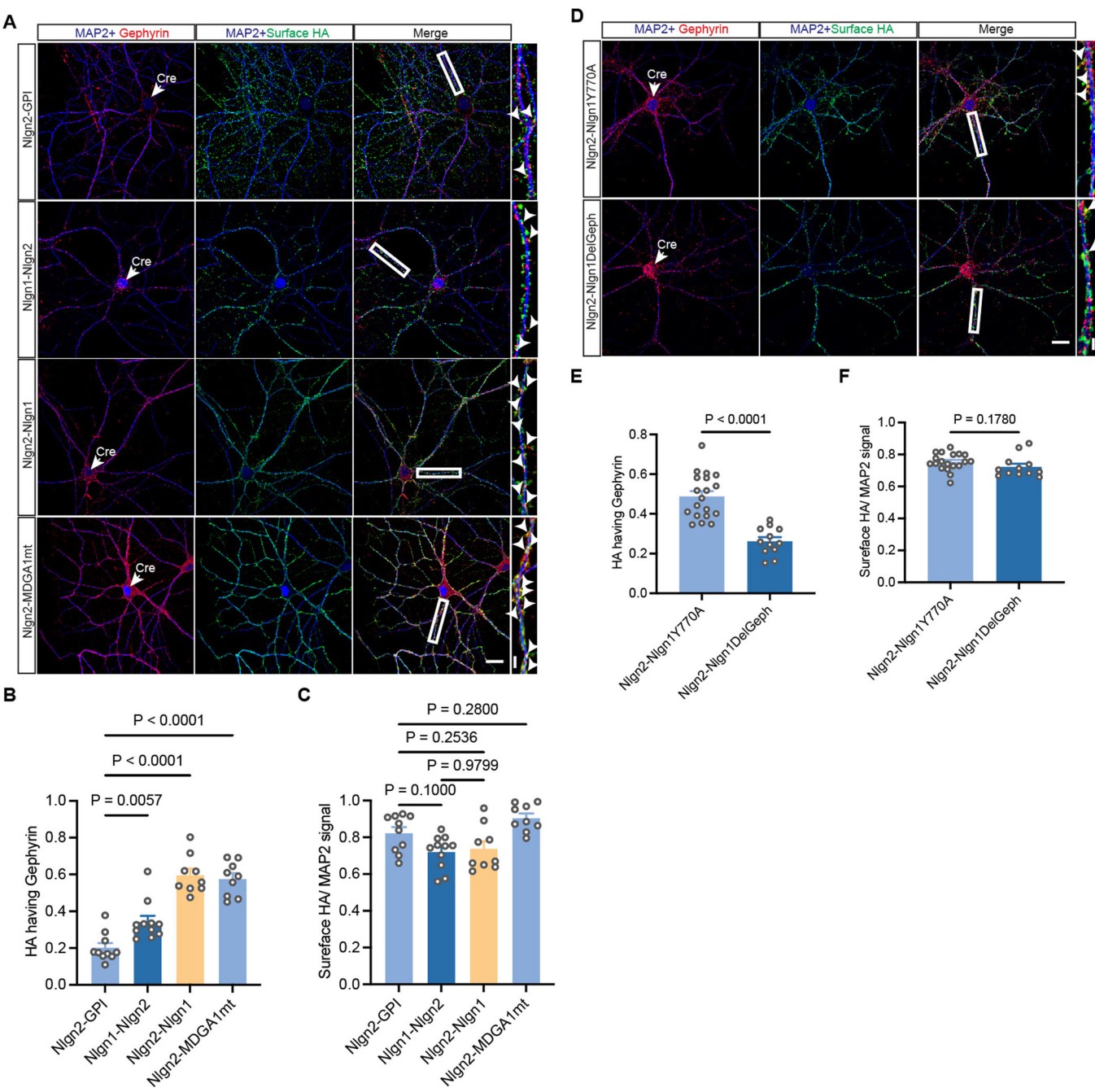

**Figure EV2. Nlgn2 is specifically localized on inhibitory synapses and is determined by the extracellular sequence of Nlgn2 and cytoplasmic gephyrin binding motif is required whereas tyrosine phosphorylation is not (related to Figs. 2, 3 and 5).**

(A) Representative image from DIV14-16 cultured Nlgn1234 conditional knockout mice neurons, infected with Cre (blue) and Nlgn2-GPI, Nlgn1-Nlgn2, Nlgn2-Nlgn1, and Nlgn2-MDGA1mt (from top to bottom). The neurons were labeled with antibodies to Gephyrin (red), HA (green), and MAP2 (blue). Scale bar: 20 μm. The right panels show an enlarged box area (arrowheads indicate HA puncta overlapped with Gephyrin puncta). Scale bar: 5 μm. (B) Summary graph of the HA-Gephyrin overlap percentage in Nlgn2-GPI, Nlgn1-Nlgn2, Nlgn2-Nlgn1, and Nlgn2-MDGA1mt conditions. (C) Summary graph of the surface levels of HA-tagged Nlgn forms relative to MAP2 signal. (B, C) (Bar and line graphs indicate mean ± SEM; numbers of cells/experiments = 10/3, 11/3, 9/3, and 9/3 for each column, left to right. Statistical significance was assessed by one-way ANOVA with post hoc Dunnett's Multiple comparisons, Nonsignificant $p > 0.05$; **$p < 0.01$; ****$p < 0.0001$). (D) Representative image from DIV14-16 cultured Nlgn1234 conditional knockout mice neurons, infected with Cre (blue) and Nlgn2-Nlgn1Y770A and Nlgn2-Nlgn1DelGeph. The neurons were labeled with antibodies to Gephyrin (red), HA (green), and MAP2 (blue). Scale bar: 20 μm. The right panels show an enlarged box area (arrowheads indicate HA puncta overlapped with Gephyrin puncta). Scale bar: 5 μm. (E) Summary graph of the HA-Gephyrin overlap percentage in Nlgn2-Nlgn1Y770A and Nlgn2-Nlgn1DelGeph conditions. (F) Summary graph of the surface levels of HA-tagged Nlgn2-Nlgn1Y770A and Nlgn2-Nlgn1DelGeph relative to MAP2 signal. (E, F) (Bar and line graphs indicate mean ± SEM; numbers of cells/experiments = 19/3 and 12/3 for each column, left to right. Statistical significance was assessed by one-way ANOVA with post hoc Dunnett's Multiple comparisons, Nonsignificant $p > 0.05$; ****$p < 0.0001$). Source data are available online for this figure.

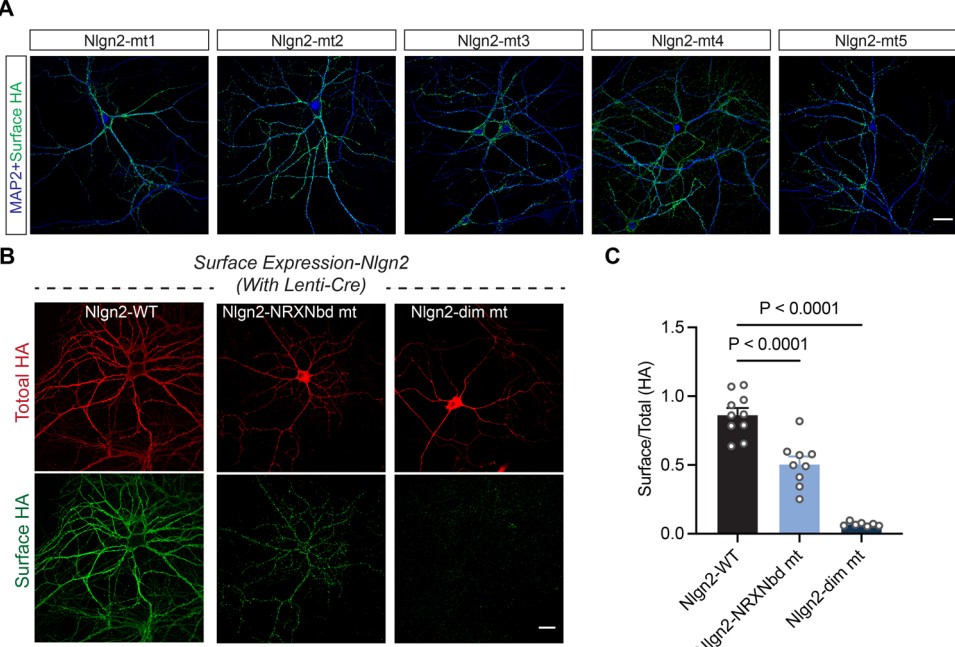

**Figure EV3.   All Nlgn2 intracellular domain truncation constructs can properly traffic to the neuron membrane surface, but Nlgn2-NRXNmt and Nlgn2-dimmt constructs traffic to the neuron membrane surface signification decrease compared to the Nlgn2-WT construct (related to Fig. 4).**

(**A**) Representative image from DIV14-16 cultured Nlgn1234 conditional knockout mice neurons, infected with Cre (blue) and Nlgn2-mt1, Nlgn2-mt2, Nlgn2-mt3, Nlgn2-mt4 and Nlgn2-mt5 (from left to right). The neurons were labeled with antibodies to HA (green) and MAP2 (blue). Scale bar: 20 μm. (**B**) Representative image from DIV14-16 cultured Nlgn1234 conditional knockout mice neurons, infected with Nlgn2-WT, Nlgn2-NRXNmt, and Nlgn2-dimmt. The neurons were labeled with antibodies to total HA (red) and surface HA (green). Scale bar: 20 μm. (**C**) Summary graph of the surface HA/total HA in Nlgn2-WT, Nlgn2-NRXNmt, and Nlgn2-dimmt conditions (Bar and line graphs indicate mean ± SEM. numbers of cells/experiments = 10/3, 9/3 and 7/3 for each column, left to right. Statistical significance was assessed by one-way ANOVA with post hoc Dunnett's Multiple comparisons, ****$p < 0.0001$). Source data are available online for this figure.

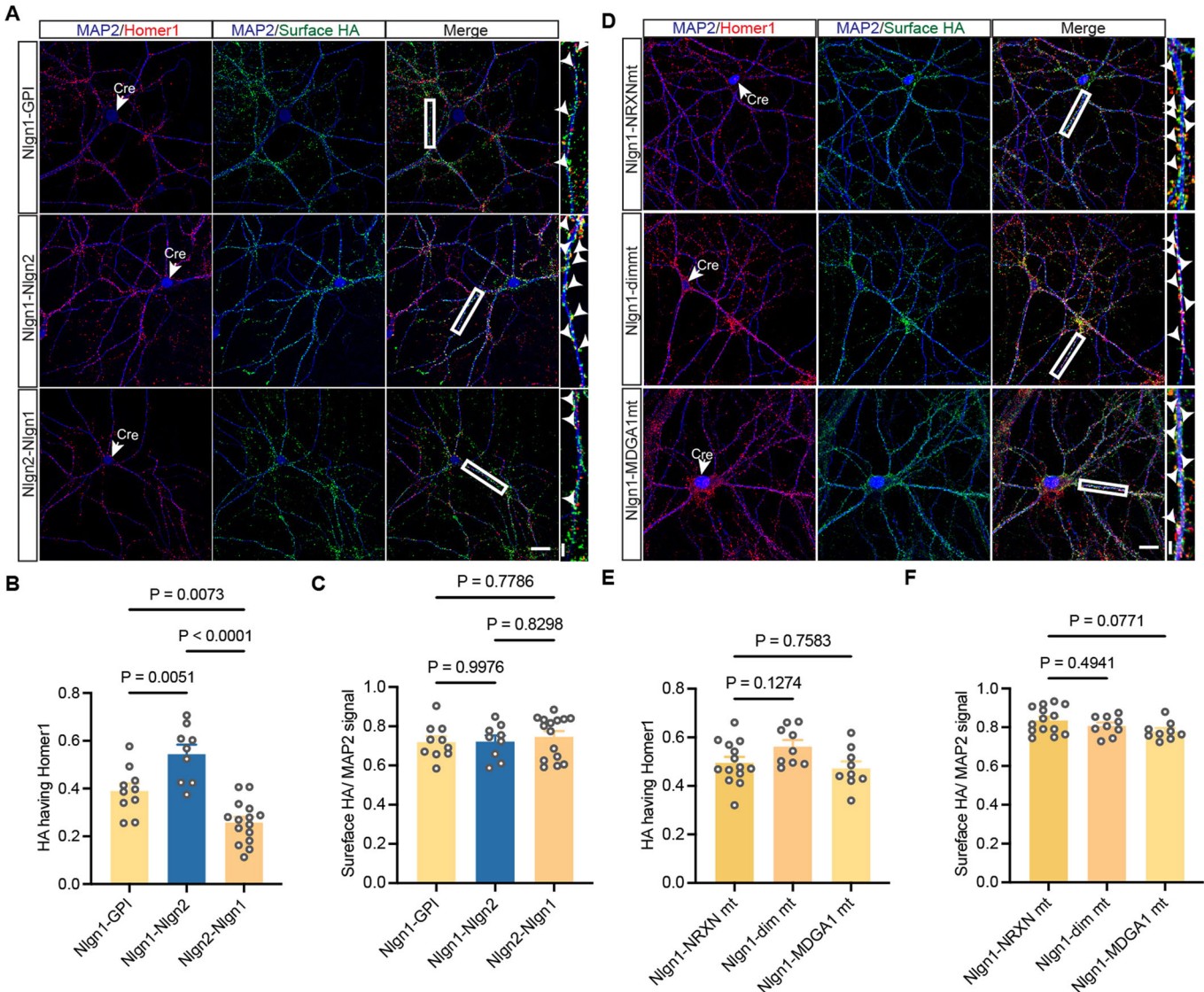

**Figure EV4.  Nlgn1 is specifically localized on excitatory synapses is determined by the extracellular sequence of Nlgn1, and does not require Neurexin-binding, MDGA-binding, or dimerization (related to Figs. 6 and 7).**

(**A**) Representative image from DIV14-16 cultured Nlgn1234 conditional knockout mice neurons, infected with Cre (blue) and Nlgn1-GPI, Nlgn1-Nlgn2, and Nlgn2-Nlgn1. The neurons were labeled with antibodies to Homer1 (red), HA (green), and MAP2 (blue). Scale bar: 20 μm. The right panels show an enlarged box area (arrowheads indicate HA puncta overlapped with Homer1 puncta). Scale bar: 5 μm. (**B**) Summary graph of the HA-Homer1 overlap percentage in Nlgn1-GPI, Nlgn1-Nlgn2, and Nlgn2-Nlgn1 conditions. (**C**) Summary graph of the surface levels of HA-tagged Nlgn1 forms relative to MAP2 signal. (**B, C**) (Bar and line graphs indicate mean ± SEM; numbers of cells/experiments = 10/3, 9/3, and 15/3 for each column, left to right. Statistical significance was assessed by one-way ANOVA with post hoc Dunnett's Multiple comparisons, Nonsignificant $p > 0.05$; **$p < 0.01$; ****$p < 0.0001$). (**D**) Representative image from DIV14-16 cultured Nlgn1234 conditional knockout mice neurons, infected with Cre (blue) and Nlgn1-NRXNmt, Nlgn1-dimmt, and Nlgn1-MDGA1mt. The neurons were labeled with antibodies to Homer1 (red), HA (green), and MAP2 (blue). Scale bar: 20 μm. The right panels show an enlarged box area (arrowheads indicate HA puncta overlapped with Homer1 puncta). Scale bar: 5 μm. (**E**) Summary graph of the HA-Homer1 overlap percentage in Nlgn1-NRXNmt, Nlgn1-dimmt, and Nlgn1-MDGA1mt conditions. (**F**) Summary graph of the surface levels of HA-tagged Nlgn1 forms relative to MAP2 signal. (**E, F**) (Bar and line graphs indicate mean ± SEM; numbers of cells/experiments = 14/3, 9/3, and 9/3 for each column, left to right. Statistical significance was assessed by one-way ANOVA with post hoc Dunnett's Multiple comparisons, Nonsignificant $p > 0.05$). Source data are available online for this figure.

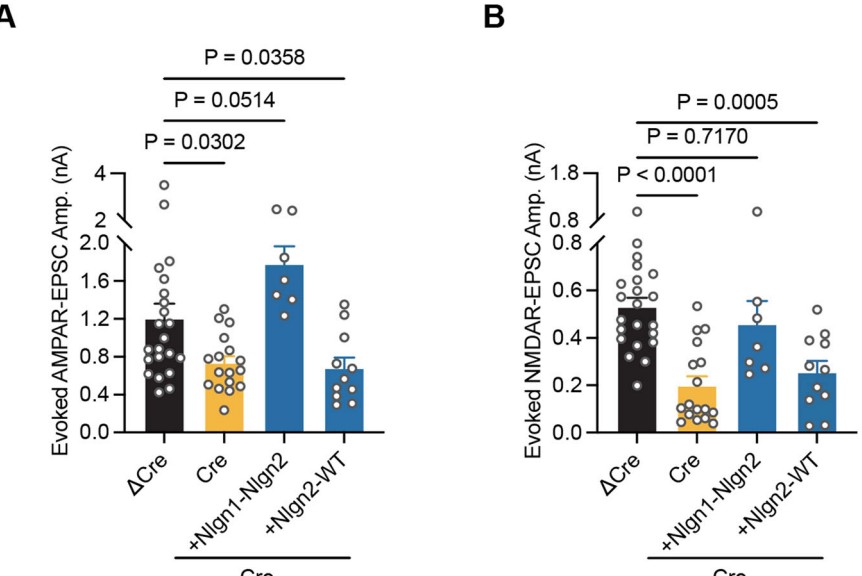

**Figure EV5. Nlgn1-Nlgn2 is sufficient for the glutamatergic synaptic transmission function of Nlgn1, but Nlgn2-WT doesn't (related to Fig. 6).**

(A) Summary graph of evoked AMPAR-EPSC amplitude in all conditions (Bar and line graphs indicate mean ± SEM; numbers of cells/experiments = 22/4, 17/4, 22/4, 7/2, and 11/3 for each column, left to right). Nonsignificant $p > 0.05$; *$p < 0.05$, one-way ANOVA with post hoc Dunnett's Multiple comparisons. Nonsignificant relations are indicated as ns. Note that we rescued Nlgn1-Nlgn2 and Nlgn2-WT here from the same batches of Fig. 6, so the ΔCre and Cre data here are the same as in Fig. 6. (B) Summary graph of evoked NMDAR-EPSC amplitude in all conditions (Bar and line graphs indicate mean ± SEM; numbers of cells/experiments = 22/4, 17/4, 22/4, 7/2, and 11/3 for each column, left to right). Nonsignificant $p > 0.05$; ***$p < 0.001$; ****$p < 0.0001$, one-way ANOVA with post hoc Dunnett's Multiple comparisons. Nonsignificant relations are indicated as ns. Note that we rescued Nlgn1-Nlgn2 and Nlgn2-WT here from the same batches of Fig. 6, so the ΔCre and Cre data here are the same as in Fig. 6. Source data are available online for this figure.

