## [Peer Review File · EMBO Reports]

Distinct mechanisms control the specific synaptic functions of Neuroligin 1 and Neuroligin 2

Jinzhao Wang, Thomas Südhof, and Marius Wernig

Corresponding author(s): Marius Wernig (wernig@stanford.edu), Thomas Südhof (tcs1@stanford.edu)

Review Timeline:

Submission Date:	23rd Dec 23
Editorial Decision:	20th Feb 24
Revision Received:	23rd Jun 24
Editorial Decision:	25th Jul 24
Revision Received:	16th Sep 24
Accepted:	26th Sep 24

Transaction Report:

Dear Dr. Wernig

Thank you for the submission of your research manuscript to our journal. We have now received the full set of referee reports that is copied below.

As you will see, while the referees acknowledge that the findings are potentially interesting, they all point out that significant revisions are required and that the data need to be strengthened before the study can be considered for publication here. The referees raise concerns about the localization and expression levels of the different Nlgn constructs used, missing controls, and the conclusiveness of the data on the requirement of extracellular and intracellular Nlgn domains.

From these comments it is clear that publication of the manuscript in our journal cannot be considered at this stage. On the other hand, given the potential interest of your findings, I would like to give you the opportunity to address the concerns and would be willing to consider a revised manuscript with the understanding that the referee concerns must be fully addressed and their suggestions (as detailed above and in their reports) taken on board.

Should you decide to embark on such a revision, acceptance of the manuscript will depend on a positive outcome of a second round of review and I should also remind you that it is EMBO reports policy to allow a single round of revision only and that, therefore, acceptance or rejection of the manuscript will depend on the completeness of your responses included in the next, final version of the manuscript.

We realize that it is difficult to revise to a specific deadline. In the interest of protecting the conceptual advance provided by the work, we recommend a revision within 3 months (May 20). Please discuss the revision progress ahead of this time with the editor if you require more time to complete the revisions.

I am also happy to discuss the revision further via e-mail or a video call, if you wish.

*****IMPORTANT NOTE:

We perform an initial quality control of all revised manuscripts before re-review. Your manuscript will FAIL this control and the handling will be delayed IN CASE the following APPLIES:

- 1) A data availability section providing access to data deposited in public databases is missing. If you have not deposited any data, please add a sentence to the data availability section that explains that.
- 2) Your manuscript contains statistics and error bars based on $n=2$. Please use scatter blots in these cases. No statistics should be calculated if $n=2$.

When submitting your revised manuscript, please carefully review the instructions that follow below. Failure to include requested items will delay the evaluation of your revision. *****

2) individual production quality figure files as .eps, .tif, .jpg (one file per figure). Please download our Figure Preparation Guidelines (figure preparation pdf) from our Author Guidelines pages <https://www.embopress.org/page/journal/14693178/authorguide> for more info on how to prepare your figures.

4) a complete author checklist, which you can download from our author guidelines (<<https://www.embopress.org/page/journal/14693178/authorguide>>). Please insert information in the checklist that is also reflected in the manuscript. The completed author checklist will also be part of the RPF.

5) Please note that all corresponding authors are required to supply an ORCID ID for their name upon submission of a revised manuscript (<<https://orcid.org/>>). Please find instructions on how to link your ORCID ID to your account in our manuscript tracking system in our Author guidelines (<<https://www.embopress.org/page/journal/14693178/authorguide#authorshipguidelines>>)

6) We replaced Supplementary Information with Expanded View (EV) Figures and Tables that are collapsible/expandable online. A maximum of 5 EV Figures can be typeset. EV Figures should be cited as "Figure EV1, Figure EV2" etc... in the text and their respective legends should be included in the main text after the legends of regular figures.

7) Please note that a Data Availability section at the end of Materials and Methods is now mandatory. In case you have no data that requires deposition in a public database, please state so instead of refereeing to the database. See also <<https://www.embopress.org/page/journal/14693178/authorguide#dataavailability>>. Please note that the Data Availability Section is restricted to new primary data that are part of this study.

Additional information on source data and instruction on how to label the files are available <<https://www.embopress.org/page/journal/14693178/authorguide#sourcedata>>.

10) Figure legends and data quantification:
The following points must be specified in each figure legend:

- the name of the statistical test used to generate error bars and P values,
 - the number (n) of independent experiments (please specify technical or biological replicates) underlying each data point,
 - the nature of the bars and error bars (s.d., s.e.m.)
- If the data are obtained from n {less than or equal to} 5, show the individual data points in addition to the SD or SEM.
- If the data are obtained from n {less than or equal to} 2, use scatter blots showing the individual data points.

See also the guidelines for figure legend preparation:
<https://www.embopress.org/page/journal/14693178/authorguide#figureformat>

11) Our journal encourages inclusion of *data citations in the reference list* to directly cite datasets that were re-used and obtained from public databases. Data citations in the article text are distinct from normal bibliographical citations and should directly link to the database records from which the data can be accessed. In the main text, data citations are formatted as follows: "Data ref: Smith et al, 2001" or "Data ref: NCBI Sequence Read Archive PRJNA342805, 2017". In the Reference list, data citations must be labeled with "[DATASET]". A data reference must provide the database name, accession number/identifiers and a resolvable link to the landing page from which the data can be accessed at the end of the reference. Further instructions are available at <<https://www.embopress.org/page/journal/14693178/authorguide#referencesformat>>.

12) All Materials and Methods need to be described in the main text. We would encourage you to use 'Structured Methods', our new Materials and Methods format. According to this format, the Materials and Methods section should include a Reagents and Tools Table (listing key reagents, experimental models, software and relevant equipment and including their sources and relevant identifiers) followed by a Methods and Protocols section in which we encourage the authors to describe their methods using a step-by-step protocol format with bullet points, to facilitate the adoption of the methodologies across labs. More information on how to adhere to this format as well as downloadable templates (.doc or .xls) for the Reagents and Tools Table can be found in our author guidelines: <<https://www.embopress.org/page/journal/14693178/authorguide#manuscriptpreparation>>.

An example of a Method paper with Structured Methods can be found here:
<<https://www.embopress.org/doi/10.15252/msb.20178071>>.

13) As part of the EMBO publication's Transparent Editorial Process, EMBO Reports publishes online a Review Process File to accompany accepted manuscripts. This File will be published in conjunction with your paper and will include the referee reports, your point-by-point response and all pertinent correspondence relating to the manuscript.

Yours sincerely,

Referee #1:

The authors analyzed the function and localization of two synaptic adhesion molecules, neuroligin 1 and neuroligin 2. Over the last two decades, these proteins have been shown to participate in the formation of excitatory synapses (neuroligin 1) and inhibitory synapses (neuroligin 2), with much of the insight arising from the laboratory submitting this manuscript. Here, the authors rely on Neuroligin1,2,3,4 quadruple KO neurons to analyze important details of the neuroligin 1 and 2 function. They conclude that specific domains of these molecules are essential for their synaptic location and function, while others are dispensable, at least in part.

The manuscript adds new evidence to our knowledge of neuroligin function. The experiments are generally well performed and interpreted. Nonetheless, the manuscript could be strongly improved by addressing a number of experimental and/or interpretation issues:

- 1) The number of cells analyzed in some of the experiments seems rather low (for example, see Figure 1D and 1F). This reviewer assumes that this is due to difficulties in obtaining the quadruple KO cells. The authors should comment on this in their methods.
- 2) The difference in the colocalization of specific neuroligins to inhibitory or excitatory synapses, as observed in Figure 1, is rather low, conflicting with the claims of the authors (and of the field) relating to the specific function of these molecules in excitatory or inhibitory synapses. The authors claim that this may be due to the low resolution of the imaging method used. For a clearer perspective, they should test this claim, by analyzing the colocalization of the synapse markers used, Homer1 (excitatory synapse marker) and Gephyrin (inhibitory synapse marker). Should these markers show a significant colocalization, then the claim of the authors is fully valid. Otherwise, the authors should discuss and interpret the presence of high levels of neuroligins at the "wrong" synapses.
- 3) In Figure 2, the Nlgn2-GPI does not restore function, presumably because it is not localized properly to synapses. The authors should explain this situation more clearly.
- 4) To strengthen the claims made in Figure 3, the authors should add an experiment using constructs expressing the Nlgn2 cytosolic domain, instead of Nlgn1 cytosolic domain.
- 5) In the majority of the Expanded View Figures, the authors show the presence of different constructs at synapses as the % overlap of HA puncta with either Homer1 or Gephyrin puncta. This analysis does not consider the levels of expression of the different constructs. The authors should quantify this, in fluorescence intensity units, for the different constructs.
- 6) In regards to Figures EV1 and EV5, the GPI-anchored Nlgn1 or Nlgn2 construct has a spotty appearance, similar to the other constructs. This is surprising, to some extent, as GPI-anchored proteins often show wide, diffuse patterns. Would stronger

fixation result in a more diffuse pattern?

7) For completeness, the authors should investigate the Nlgn1-Nlgn2 construct, described in Fig. 5, also in Fig. 6, in the analysis of the AMPA and NMDA receptor responses.

Referee #2:

Summary

Wang and colleagues investigated the isoform specificity of neuroligins (Nlgns), postsynaptic cell-adhesion molecules regulating synaptic functions. Despite high homology, Nlgn1 and Nlgn2 are distinct in terms of the synaptic localization. Electrophysiological recordings from cultured hippocampal mouse neurons lacking all four Nlgns confirmed that Nlgn1 and Nlgn2 selectively regulate excitatory and inhibitory synaptic transmission, respectively, aligning with their synaptic localizations. By utilizing different chimeric Nlgn1-Nlgn2 constructs, the authors showed that extracellular domains of Nlgns determine their synapse specificity actions, while intracellular sequences were dispensable. Notably, the cytoplasmic sequences of Nlgn2, including its gephyrin-binding motif, are essential for synaptic function while such case does not exist for Nlgn1. In summary, this paper highlighted that although the extracellular sequences dictate excitatory or inhibitory synapse specificity, distinct intracellular mechanisms underlie the normal synaptic connections enabled by Nlgn1 and Nlgn2. Overall, the data are extensive and systematic; issue here is how the paper is conclusive and conceptually advanced. I have some reservations and many data should be further elaborated for consideration in the EMBO Reports.

Major comments

1. Authors claimed that the extracellular domains of Nlgns determine synaptic specificity. However, the following examples do not clearly support the authors' proposition. Data presented in EV1, EV2 and EV5 also support the notion that the intracellular sequences of Nlgns contribute to synaptic specificity. Nlgn1-GPI and Nlgn2-GPI exhibit lower 'synaptic specificity', indicating the contribution of the intracellular sequences.
2. There are a number of figures, where WT control is missing (Figs. 3, 4, 7, EV1, EV2, EV4 and EV5). The WT control should be included for a clear-cut interpretation of the respective results. In addition, the various constructs generated in this study should exhibit comparable expression/surface expression levels as wild-type control - the relevant results should be provided.
3. In Figure 5, the authors concluded that interaction of Nlgn2 with MDGA1 is not involved in mediating the exclusive specificity of Nlgn2 at inhibitory synapses. The straightforward interpretation should be that MDGA1 does not suppress the Nlgn2 action, not necessarily suggesting that MDGA1 does not mediate the synaptic specificity of Nlgn2. The author should also be attentive to a recent work showing amyloid precursor protein is a target for Nlgn2 at the hippocampal synapses (Kim 2022 PNAS). The relevant description should be added because this aligned well with the authors' findings here.
4. In Figure 7: similar sets of experiments should be performed for Nlgn2-NRNmut, Nlgn2-dimmt, and Nlgn2-MDGA1 mt to ensure that findings here can be convincingly concluded.
5. Discussion (line 19-23): the authors stated that "while chimeric Nlgn1-Nlgn2 molecules mediate glutamatergic, but not GABAergic, synaptic transmission". However, the only evidence authors provided here is that Nlgn1-Nlgn2 failed to rescue GABAAR-mediated IPSCs presented in Figures 5B-C. I think that the authors should show that Nlgn1-Nlgn2 can rescue AMPAR- and NMDAR-EPSCs as they have done so for Nlgn1-GPI and Nlgn1-WT in Figures 6B-D.

Minor comments

1. Figure 1: HA-Nlgn1 and HA-Nlgn2 do not have comparable surface expression level when measuring colocalization with Homer1 and gephyrin, respectively. For example, surface HA-Nlgn2 levels appear to be less than surface HA-Nlgn1 (panel C). The opposite is true in panel E. These representative images concern the reviewer to believe that these analyses are not reliable.
2. Figure 1: Please provide Nlgn1/2/3/4 KO validation results.
3. Figure 2: do Nlgn2-WT, Nlgn2-GPI and Nlgn2-Nlgn1 exhibit comparable expression and surface expression levels?
4. Which gephyrin-binding sequence is correct in Figure 3A and B? There is a clear disparity.
5. Figure 3: authors need to show that Nlgn2-Nlgn1 Y770A and Nlgn2-Nlgn1 DelGeph do not retain the binding activity to gephyrin using reliable biochemistry experiments.
6. Figure 4: The rationale to set boundaries for cutting intracellular sequence of Nlgns should be provided in detail. In addition, more detailed explanation/descriptions should be provided for enhanced readership (e.g., very difficult to follow-up a 21 amino acid-long sequence as presented in the text...).
7. In EV1: gephyrin immunofluorescence intensity vary in the representative images, particularly those for Nlgn2-Nlgn1 and Nlgn2-MDGA1mut, which show clearly better colocalization (panel B). Moreover, surface expression pattern of Nlgn2-MDGA1 mut is quite different from the other Nlgn variants.

Referee #3:

Wang et al, examine the molecular mechanisms by which Neuroligin 1 and Neuroligin 2 regulate excitatory and inhibitory

synapses in hippocampal neurons. To specifically address this question, the authors generated a quadruple conditional KO mouse model where Nlgn1, 2, 3 and 4 are missing. Neurons from these mice were cultured in vitro and diverse constructs for Nlgn1 and 2 were expressed to examine the impact on the surface levels of these neuroligins and their impact on excitatory or inhibitory synapses using electrophysiological recording and cell biology approaches. They also used specific mutant of Nlgn where the intracellular domains were exchanged between Nlgn1 and Nlgn2, or the intracellular domain carrying specific mutations that affect the interaction with neuexin or also with MDGAs, which are cell-adhesion molecules that compete with neuexins.

These studies provide new and unexpected results. First the function of Nlgn2 at inhibitory synapses requires its intracellular domain. However, this is not the same with Nlgn1. Second, a mutant for Nlgn2 (Y770), which disrupt the binding to gephyrin is fully functional raising the interesting question as to how Nlgn2 regulates inhibitory synapses. Third, the extracellular domain but not the intracellular domain of Nlgn1 is required for its specificity to excitatory synapses. In summary, these two Nlgn uses different mechanisms to obtain specificity. These new findings raise questions about previous proposed models on how Nlgn regulates excitatory and inhibitory synapses. These models will need to be re-evaluated more carefully. Overall, the results presented in this manuscript will be a great value to the community studying the assembly and function of excitatory and inhibitory synapses and in particular the role of Nlgn and neuexin function in the nervous system. However, there are a number of important issues that the authors need to address before this paper is considered further for publication in EMBO Reports.

- 1) A major deficiency of the manuscript is the lack of individual points in all the graphs presented. The graphs should present all the data points obtain from individual Ns.
- 2) In Figure EV4 B, the authors present a graph depicting the levels of surface Nlgn2 WT, Nlgn2-NRXNmt and Nlgn2-dimmt normalised to total. It is surprising that more than 40% of WT Nlgn is present at the cell surface. This is a very high value for any surface protein. Could the authors explain this result? How do they determine this? Based on just the fluorescence levels? The authors should also should evaluate this by biochemical methods.
- 3) Figure EV5B: is the difference between Nlgn1-GPI statistically different from the Nlgn2-Nlgn1?
- 4) The authors should explain more the implications for their work in relation to published literature on neuroligins. This seems very minimal in the current discussion.

Minor comments:

- 1) the n is missing after in Nlgn2 in Figure EV5B.
- 2) The list of references is corrupted. Page 2 of the references starts with reference 1 again like page 1 and the same for subsequent reference pages.
- 3) The figures were not properly labelled.

Referee #1:

The authors analyzed the function and localization of two synaptic adhesion molecules, neuroligin 1 and neuroligin 2. Over the last two decades, these proteins have been shown to participate in the formation of excitatory synapses (neuroligin 1) and inhibitory synapses (neuroligin 2), with much of the insight arising from the laboratory submitting this manuscript. Here, the authors rely on Neuroligin1,2,3,4 quadruple KO neurons to analyze important details of the neuroligin 1 and 2 function. They conclude that specific domains of these molecules are essential for their synaptic location and function, while others are dispensable, at least in part.

The manuscript adds new evidence to our knowledge of neuroligin function. The experiments are generally well performed and interpreted. Nonetheless, the manuscript could be strongly improved by addressing a number of experimental and/or interpretation issues:

1) The number of cells analyzed in some of the experiments seems rather low (for example, see Figure 1D and 1F). This reviewer assumes that this is due to difficulties in obtaining the quadruple KO cells. The authors should comment on this in their methods.

Response: We acknowledge the concern regarding the relatively low number of cells analyzed in certain experiments, as highlighted in Figure 1D and 1F. Given the effect size and the consistency of results we feel that the number of cells analyzed is sufficient but if recommended, we would be happy to measure more cells (which would take approximately 3-4 months). We have now included a statement in the Methods section to clarify this issue. Please see page 16 paragraph 1 the blue highlight of our revised manuscript.

2) The difference in the colocalization of specific neuroligins to inhibitory or excitatory synapses, as observed in Figure 1, is rather low, conflicting with the claims of the authors (and of the field) relating to the specific function of these molecules in excitatory or inhibitory synapses. The authors claim that this may be due to the low resolution of the imaging method used. For a clearer perspective, they should test this claim, by analyzing the colocalization of the synapse markers used, Homer1 (excitatory synapse marker) and Gephyrin (inhibitory synapse marker). Should these markers show a significant colocalization, then the claim of the authors is fully valid. Otherwise, the authors should discuss and interpret the presence of high levels of neuroligins at the "wrong" synapses.

Response: Thank you for this important feedback regarding Figure 1 of our manuscript. We agree that co-staining Homer1 with Gephyrin will help address this question and have now performed this experiment. However, we would like to note that at the level of a confocal analysis, the background of a positive colocalization with the 'wrong' synapse is high because the imaging resolution is not sufficient for separating closely spaced synapses and because thresholding has a huge effect on apparent colocalizations. Please see page 5 highlight part and Figure EV1 C and D.

3) In Figure 2, the Nlgn2-GPI does not restore function, presumably because it is not localized properly to synapses. The authors should explain this situation more clearly.

Response: Thanks for this careful attention to this data point. Following this comment, we agree that the lack of functional restoration by Nlgn2-GPI may indeed be attributed to its improper localization to synapses. We will revise the corresponding section of the manuscript to reflect this possibility. Specifically, we will discuss the potential factors contributing to the improper localization of Nlgn2-GPI, such as the absence of synaptic targeting motifs in the GPI-anchor domain and the impact of membrane anchoring on protein trafficking and synaptic recruitment. Please see page 6 paragraph 2 the blue highlight of our revised manuscript.

4) To strengthen the claims made in Figure 3, the authors should add an experiment using constructs expressing the Nlgn2 cytosolic domain, instead of Nlgn1 cytosolic domain.

Response: We agree that it would be more elegant to use Nlgn2's intracellular domain for these experiments. However, we would like to respectfully argue that the results using Nlgn1's intracellular domain are informative and given the amount of effort needed to repeat this entire set of experiments -

the potential gain in knowledge rather limited. Please note that the intracellular domain between the two molecules is well conserved, in particular the Gephyrin binding domain and, most importantly, the WT Nlgn1 intracellular domain perfectly rescues Nlgn2 function.

5) In the majority of the Expanded View Figures, the authors show the presence of different constructs at synapses as the % overlap of HA puncta with either Homer1 or Gephyrin puncta. This analysis does not consider the levels of expression of the different constructs. The authors should quantify this, in fluorescence intensity units, for the different constructs.

Response: We agree that quantifying the fluorescence intensity units of different constructs would provide valuable additional information. We revised the analysis in the Expanded View Figures to include quantification of fluorescence intensity units for the different constructs. Please see Figure EV2 C&F, Figure EV3 C and Figure EV4 C&F.

6) In regard to Figures EV1 and EV5, the GPI-anchored Nlgn1 or Nlgn2 construct has a spotty appearance, similar to the other constructs. This is surprising, to some extent, as GPI-anchored proteins often show wide, diffuse patterns. Would stronger fixation result in a more diffuse pattern?

Response: We would like to thank the reviewer for this insightful comment. Indeed, one would expect that GPI-anchored proteins should generally be more diffuse. Obviously, the extracellular Nlgn domains contribute to the dotted expression pattern. We have included this insight into our revised manuscript. Just to clarify: the Nlgn constructs were all incubated live with an anti-HA monoclonal antibody and then fixed and further processed. Please see page 15 paragraph 3 the blue highlight of our revised manuscript.

7) For completeness, the authors should investigate the Nlgn1-Nlgn2 construct, described in Fig. 5, also in Fig. 6, in the analysis of the AMPA and NMDA receptor responses.

Response: This is an excellent suggestion and we have performed the requested experiments. These data are now shown in Figure EV5 in our revised manuscript. Thank you for your valuable feedback.

Referee #2:

Summary

Wang and colleagues investigated the isoform specificity of neuroligins (Nlgn), postsynaptic cell-adhesion molecules regulating synaptic functions. Despite high homology, Nlgn1 and Nlgn2 are distinct in terms of the synaptic localization. Electrophysiological recordings from cultured hippocampal mouse neurons lacking all four Nlgn confirmed that Nlgn1 and Nlgn2 selectively regulate excitatory and inhibitory synaptic transmission, respectively, aligning with their synaptic localizations. By utilizing different chimeric Nlgn1-Nlgn2 constructs, the authors showed that extracellular domains of Nlgn determine their synapse specificity actions, while intracellular sequences were dispensable. Notably, the cytoplasmic sequences of Nlgn2, including its gephyrin-binding motif, are essential for synaptic function while such case does not exist for Nlgn1. In summary, this paper highlighted that although the extracellular sequences dictate excitatory or inhibitory synapse specificity, distinct intracellular mechanisms underlie the normal synaptic connections enabled by Nlgn1 and Nlgn2. Overall, the data are extensive and systematic; issue here is how the paper is conclusive and conceptually advanced. I have some reservations and many data should be further elaborated for consideration in the EMBO Reports.

We are grateful for these positive remarks and thank the reviewer for sharing his/her concerns that we hope have addressed as well as possible.

Major comments

1. Authors claimed that the extracellular domains of Nlgn determine synaptic specificity. However, the following examples do not clearly support the authors' proposition. Data presented in EV1, EV2 and EV5 also support the notion that the intracellular sequences of Nlgn contribute to synaptic specificity. Nlgn1-GPI and Nlgn2-GPI exhibit lower 'synaptic specificity', indicating the contribution of the intracellular sequences.

Response: We apologize if our description may have been unclear. As the reviewer points out the intracellular domain of Nlgn2 (but intriguingly not of Nlgn1) is necessary for synaptic localization and its "specific" recruitment to synapses. The specificity for glutamatergic vs. GABAergic synapses, however, is dictated by the Nlgn1, and Nlgn2 extracellular domains, respectively. We have edited the text in the revised manuscript to clarify this point. Please see page 2 paragraph 1 and page 12 paragraph 3 the blue highlights of our revised manuscript.

2. There are a number of figures, where WT control is missing (Figs. 3, 4, 7, EV1, EV2, EV4 and EV5). The WT control should be included for a clear-cut interpretation of the respective results. In addition, the various constructs generated in this study should exhibit comparable expression/surface expression levels as wild-type control - the relevant results should be provided.

Response: It is an excellent comment that it is of utmost importance that the levels of the WT Nlgn and the mutant Nlgn versions have to be similar in order to draw solid conclusions. We have therefore performed detailed quantification of the surface expression of all constructs used in this study. These data are now shown in our new Figure EV2C and F, Figure EV4C and F in the revised manuscript. With regards to the second component of the comment, and in light of these new results, we must thoroughly apologize that we cannot exactly follow the reviewer's reasoning why the addition of the WT versions would be necessary for all these experiments. Both Nlgn1 & 2 WT constructs rescue their respective phenotypes to 100% (Figure 2 and 6). Now, thanks to the reviewer, we also know that the levels of the mutant constructs are comparable to different versions. Hence, in those cases in which the mutant Nlgn show full rescue (e.g. Nlgn2-Nlgn1Y770A in Figure 3, Nlgn2-mt1, Nlgn2-mt2 in Figure 4, Nlgn2-MDGA1mt in Figure 5) it is safe to assume that those mutations do not interfere with the function of the respective WT Nlgn, whereas the mutations that do not rescue are of functional importance. Those are the only conclusions we would like to draw from these data. Moreover, from a practical point of view, and this may not be so obvious to the reader, if we were to add WT constructs to all these experiments, we would have to essentially repeat 90% of the manuscript's experiments which would amount to a 1–2-year effort.

3. In Figure 5, the authors concluded that interaction of Nlgn2 with MDGA1 is not involved in mediating the exclusive specificity of Nlgn2 at inhibitory synapses. The straightforward interpretation should be that MDGA1 does not suppress the Nlgn2 action, not necessarily suggesting that MDGA1 does not mediate the synaptic specificity of Nlgn2. The author should also be attentive to a recent work showing amyloid precursor protein is a target for Nlgn2 at the hippocampal synapses (Kim 2022 PNAS). The relevant description should be added because this aligned well with the authors' findings here.

Response: We fully agree with this comment and have adjusted the description of the MDGA1-related data accordingly. We have also included a reference and discussion about the findings of Kim et al. (2022). Please see page 8 paragraph 3 the blue highlight of our revised manuscript.

4. In Figure 7: similar sets of experiments should be performed for Nlgn2-NRNmut, Nlgn2-dimmt, and Nlgn2-MDGA1 mt to ensure that findings here can be convincingly concluded.

Response: It is well established that Nlgn2 does not affect glutamatergic synapse function, but we agree it would add more rigor to verify this notion in the same context of Figure 7. We have therefore performed Nlgn2 rescue experiments and found that Nlgn2 does not affect glutamatergic synapse function in the same experimental system as shown for the Nlgn1 molecules. These data are now shown in Figure EV5 of our revised manuscript. Since Nlgn2WT did not have any function in this assay, we did not continue with Nlgn2 mutant constructs.

5. Discussion (line 19-23): the authors stated that "while chimeric Nlgn1-Nlgn2 molecules mediate glutamatergic, but not GABAergic, synaptic transmission". However, the only evidence authors provided here is that Nlgn1-Nlgn2 failed to rescue GABAAR-mediated IPSCs presented in Figures 5B-C. I think that the authors should show that Nlgn1-Nlgn2 can rescue AMPAR- and NMDAR-EPSCs as they have done so for Nlgn1-GPI and Nlgn1-WT in Figures 6B-D.

Response: This is an excellent suggestion and we have performed the requested experiments. These data are now shown in Figure EV5 in our revised manuscript. Thank you for your valuable feedback.

Minor comments

1. Figure 1: HA-Nlgn1 and HA-Nlgn2 do not have comparable surface expression level when measuring colocalization with Homer1 and gephyrin, respectively. For example, surface HA-Nlgn2 levels appear to be less than surface HA-Nlgn1 (panel C). The opposite is true in panel E. These representative images concern the reviewer to believe that these analyses are not reliable.

Response: We appreciate the thorough evaluation of Figure 1 and raising this concern. Upon reviewing the representative images in panels C and E of Figure 1, and going through our raw image data, we found panels C and E are derived from different batch experiments and therefore should not be compared in a quantitative manner. In fact, in our new quantitation we show that the Nlgn levels are comparable between constructs (see comment above). We have now chosen representative images from the same batch of cells and staining. Please see new Figure 1 of our revised manuscript.

2. Figure 1: Please provide Nlgn1/2/3/4 KO validation results.

Response: This is also an excellent point. We have now validated the quadruple knock-out by Western blotting. Please see the new Figure EV1A and B. And page 5 paragraph 1 the blue highlight of our revised manuscript.

3. Figure 2: do Nlgn2-WT, Nlgn2-GPI and Nlgn2-Nlgn1 exhibit comparable expression and surface expression levels?

Response: We have included quantification of the surface expression levels in the revised Figure EV2C(see also response to major comment 2).

4. Which gephyrin-binding sequence is correct in Figure 3A and B? There is a clear disparity.

Response: We apologize for labeling the gephyrin-binding sequence incorrectly, we are grateful for pointing out this error and have corrected it in the revised figure. Please see Figure 3A and B.

5. Figure 3: authors need to show that Nlgn2-Nlgn1 Y770A and Nlgn2-Nlgn1DelGeph do not retain the binding activity to gephyrin using reliable biochemistry experiments.

Response: In the Figure EV2, in line with our electrophysiology data, co-labeling with Gephyrin antibodies showed that the Nlgn2-Nlgn1 Y770A mutant displayed a greater degree of co-localization with Gephyrin than the Nlgn2-Nlgn1DelGeph mutant, we don't think that an extra biochemical experiment is required.

6. Figure 4: The rationale to set boundaries for cutting intracellular sequence of Nlgn2 should be provided in detail. In addition, more detailed explanation/descriptions should be provided for enhanced readership (e.g., very difficult to follow-up a 21 amino acid-long sequence as presented in the text...).

Response: This is a great suggestion and we have edited the text accordingly. Please see page 7 paragraph 2 the blue highlight of our revised manuscript.

7. In EV1: gephyrin immunofluorescence intensity vary in the representative images, particularly those for Nlgn2-Nlgn1 and Nlgn2-MDGA1mut, which show clearly better colocalization (panel B). Moreover, surface expression pattern of Nlgn2-MDGA1 mut is quite different from the other Nlgn variants.

Response: The variation in gephyrin immunofluorescence intensity observed in Figure EV1 (Nlgn2-GPI, Nlgn1-Nlgn2, Nlgn2-Nlgn1, and Nlgn2-MDGA1 mt) is consistent with our findings. From the representative images, it is evident that the gephyrin immunofluorescence intensity is higher in Nlgn2-Nlgn1 and Nlgn2-MDGA1 mt compared to Nlgn2-GPI and Nlgn1-Nlgn2. This observation aligns with our data showing that Nlgn2-Nlgn1 and Nlgn2-MDGA1mt can rescue inhibitory synaptic transmission through gephyrin, whereas Nlgn2-GPI and Nlgn1-Nlgn2 cannot.

Referee #3:

Wang et al, examine the molecular mechanisms by which Neuroligin 1 and Neuroligin 2 regulate excitatory and inhibitory synapses in hippocampal neurons. To specifically address this question, the authors generated a quadruple conditional KO mouse model where Nlgn1, 2, 3 and 4 are missing. Neurons from these mice were cultured in vitro and diverse constructs for Nlgn1 and 2 were expressed to examine the impact on the surface levels of these neuroligins and their impact on excitatory or inhibitory synapses using electrophysiological recording and cell biology approaches. They also used specific mutant of Nlgn where the intracellular domains were exchanged between Nlgn1 and Nlgn2, or the intracellular domain carrying specific mutations that affect the interaction with neurexin or also with MDGAs, which are cell-adhesion molecules that compete with neurexins.

These studies provide new and unexpected results. First the function of Nlgn2 at inhibitory synapses requires its intracellular domain. However, this is not the same with Nlgn1. Second, a mutant for Nlgn2 (Y770), which disrupt the binding to gephyrin is fully functional raising the interesting question as to how Nlgn2 regulates inhibitory synapses. Third, the extracellular domain but not the intracellular domain of Nlgn1 is required for its specificity to excitatory synapses. In summary, these two Nlgn uses different mechanisms to obtain specificity. These new findings raise questions about previous proposed models on how Nlgn regulates excitatory and inhibitory synapses. These models will need to be re-evaluated more carefully. Overall, the results presented in this manuscript will be a great value to the community studying the assembly and function of excitatory and inhibitory synapses and in particular the role of Nlgn and neurexin function in the nervous system. However, there are a number of important issues that the authors need to address before this paper is considered further for publication in EMBO Reports.

1) A major deficiency of the manuscript is the lack of individual points in all the graphs presented. The graphs should present all the data points obtain from individual Ns.

Response: This is an important point. Our revised figures now all contain the individual data points as requested.

2) In Figure EV4 B, the authors present a graph depicting the levels of surface Nlgn2 WT, Nlgn2-NRXNmt and Nlgn2-dimnt normalized to total. It is surprising that more than 40% of WT Nlgn is present at the cell surface. This is a very high value for any surface protein. Could the authors explain this result? How do they determine this? Based on just the fluorescence levels? The authors should also evaluate this by biochemical methods.

Response: These quantifications are indeed based on quantifying fluorescence intensity by image analysis as is common practice in the field. We are not exactly clear why this value would be surprising as 40% of the total signal is only on the surface – less than half.

3) Figure EV5B: is the difference between Nlgn1-GPI statistically different from the Nlgn2-Nlgn1?

Response: We have performed a statistical analysis which showed that, indeed, Nlgn1-GPI is statistically different from Nlgn2-Nlgn1. We have now pointed that out in our revised manuscript. Please see the new Figure EV4B.

4) The authors should explain more the implications for their work in relation to published literature on neuroligins. This seems very minimal in the current discussion.

Response: We agree that further discussion on the implications of our work is warranted and have expanded the discussion accordingly in our revised manuscript. Please see page 12 paragraph 3 of our revised manuscript.

Minor comments:

1) the n is missing after in Nlg2 in Figure EV5B.

Response: We apologize for the oversight and have corrected this error. Please see the new Figure EV3C.

2) The list of references is corrupted. Page 2 of the references starts with reference 1 again like page 1 and the same for subsequent reference pages.

Response: We are grateful for spotting this mistake that we have corrected now.

3) The figures were not properly labelled.

Response: We are sorry for this oversight and have now labeled all figures. We apologize if this has caused frustration.

Dear Dr. Wernig

Thank you for the submission of your revised manuscript to EMBO reports. We have now received the full set of referee reports that is copied below.

As you will see, both referees are very positive about the study and recommend publication.

Browsing through the manuscript myself, I noticed a few editorial things that we need before we can proceed with the official acceptance of your study.

- Please provide up to 5 keywords.
- Please update the 'Conflict of interest' paragraph to our new 'Disclosure and competing interests statement'. For more information see <https://www.embopress.org/page/journal/14693178/authorguide#conflictsofinterest>
- Please add the corresponding authors' emails to the title page.
- Regarding the Author Contributions, we now use CRedit to specify the contributions of each author in the journal submission system. Therefore, please remove the Author Contributions from the manuscript file and make sure that the author contributions in our online manuscript tracking system are correct and up-to-date. The information you specified in the system will be automatically retrieved and typeset into the article. You can enter additional information in the free text box provided, if you wish.
- Please provide an ORCID ID for Dr. Wernig. You can find instructions on how to link your ORCID ID to your account in our manuscript tracking system in our Author guidelines (<https://www.embopress.org/page/journal/14693178/authorguide#authorshipguidelines>)
- Figure EV5 is not mentioned in the text. Please add a callout to Fig. EV5A and B.
- Materials and Methods should be 'Methods'
- Methods: Please add either RRID or catalogue numbers for the antibodies used.
- Methods, mouse work: Please provide the reference number for the ethics approval.
- Methods: since July 1st we require that all manuscripts contain structured methods. I know that you have submitted your manuscript before this deadline, but it would nevertheless be good, if you could convert to this format. In essence, the only thing you would need to add is a Reagents and Resources Table. You can download the template at <https://www.embopress.org/page/journal/14693178/authorguide#structuredmethods>. You do not need to change the methods as such but upload this table as file type "Reagents and Resource Table" and it will be typeset into the article. Here is an example of a manuscript with such a structured methods format: <https://www.embopress.org/doi/10.15252/msb.20178071>.
- Data availability section: Please remove the reference to the source data and instead state that no data have been deposited to public repositories. If you deposited data on BioStudies and wish to provide the URL, this can be added to the Data Availability section, but the reference to source data deposited with EMBO Press is not needed.
- Our production/data editors have asked you to clarify several points in the figure legends (see below). Please incorporate these changes in the manuscript and return the revised file with tracked changes with your final manuscript submission.

A) Figure legend text:

- Please note that the figure EV 1d is not labeled in the manuscript. This needs to be rectified.

B) Statistical test information. Only p-values that are actually shown in the figure panel(s) should (and must) be defined in the legends, all others should be removed from (or added to) the legend. Moreover, we ask for the specification of exact p-values:

- Please define the annotated p values ****/**/* as well as provide the exact p-values for the same in the legend of figure EV 1b; EV 2b, e; EV 4b; as appropriate.
- Please note that the exact p values are not provided in the legends of figures 1d, f; 2c; 3d; 4d; 5c, f; 6c-d; 7c-d; EV 3c; EV 5a-b.
- Please indicate the statistical test used for data analysis in the legends of figures EV 1b; EV 2b, e; EV 4b, e.
- Please note that in figures 1d, f; 2c; 3d; 4d; 5c, f; 6c-d; 7c-d; EV 2c, f; EV 3c; EV 4c, f; EV 5a-b; there is a mismatch between the annotated p values in the figure legend and the annotated p values in the figure file that should be corrected.

C) Replicates and error bars:

- Please note that information related to n is missing in the legends of figures EV 1b, d; EV 2b, e; EV 3c; EV 4b, e.
- Please note that the error bars are not defined in the legends of figures EV 1b, d; EV 2b, e; EV 4b, e.

C) Data presentation:

- Please note that the black and red arrowheads are not defined in the legend of figure EV 1a. This needs to be rectified.

- I introduced some minor changes to the Abstract. Please see my edited suggestion below my signature.

- Finally, EMBO Reports papers are accompanied online by

A) a short (1-2 sentences) summary of the findings and their significance,

B) 2-3 bullet points highlighting key results and

C) a schematic summary figure that provides a sketch of the major findings (not a data image).

Please provide the summary figure as a separate file in PNG or JPG format at a size of 550x300-600 pixels (width x height).

Please note that the size is rather small and that text needs to be readable at the final size. Please send us this information along with the revised manuscript.

- On a different note, I would like to alert you that EMBO Press offers a new format for a video-synopsis of work published with us, which essentially is a short, author-generated film explaining the core findings in hand drawings, and, as we believe, can be very useful to increase visibility of the work. This has proven to offer a nice opportunity for exposure i.p. for the first author(s) of the study. Please see the following link for representative examples and their integration into the article web page:

<https://www.embopress.org/doi/full/10.15252/emj.2019103932>

With kind regards,

Martina Rembold, PhD

Senior Editor

EMBO reports

Referee #1:

The authors replied thoroughly to my comments, and I am happy to suggest that the manuscript be published in its present form.

Referee #2:

In this revised manuscript, authors extensively and satisfactorily addressed all of my previous comments. I recommend its publication at EMBO reports.

Abstract

Neuroligins are postsynaptic cell-adhesion molecules that regulate synaptic function with a remarkable isoform specificity. Although Nlgn1 and Nlgn2 are highly homologous and biochemically interact with the same extra- and intracellular proteins, Nlgn1 selectively functions in excitatory synapses whereas Nlgn2 functions in inhibitory synapses. How this excitatory/ inhibitory (E/I) specificity arises is unknown. Using a comprehensive structure-function analysis, we here express wild-type and mutant neuroligins in functional rescue experiments in cultured hippocampal neurons lacking all endogenous neuroligins. Electrophysiology confirms that Nlgn1 and Nlgn2 selectively restore excitatory and inhibitory synaptic transmission, respectively, in neuroigin-deficient neurons, aligned with their synaptic localizations. Chimeric Nlgn1-Nlgn2 constructs reveal that the extracellular neuroigin domains confer synapse specificity, whereas their intracellular sequences are exchangeable. However, the cytoplasmic sequences of Nlgn2, including its Gephyrin-binding motif that is identically present in the Nlgn1, is essential for its synaptic function whereas they are dispensable for Nlgn1. These results demonstrate that although the excitatory vs. inhibitory synapse specificity of Nlgn1 and Nlgn2 are both determined by their extracellular sequences, these neuroligins enable normal synaptic connections via distinct intracellular mechanisms.

The authors have addressed all minor editorial requests.

Dr. Marius Wernig
Stanford University School of Medicine
Institute for Stem Cell Biology and Regenerative Medicine
Dept. of Pathology
1050 Arastradero Road A247
Palo Alto, CA 94304
United States

Dear Dr. Wernig,

Thank you for implementing the last minor corrections. I am very pleased to accept your manuscript for publication in the next available issue of EMBO reports. Thank you for your contribution to our journal.

Yours sincerely,
